# REVIEW, REVISE, AND LEARN: PEER-GUIDED PREFERENCE LEARNING VIA LLM SELF-CORRECTION

## ABSTRACT

Preference optimization plays a central role in achieving state-of-the-art performance in large language models (LLMs). However, preference learning requires large-scale, high-quality, or human-annotated datasets, which poses a significant challenge to the continual improvement of LLMs. We introduce **PULSE** (**P**eer-g**U**ided Preference **L**earning via LLM **SE**lf-correction), a collaborative framework of multiple LLM agents for scalable preference learning. PULSE is inspired by the academic peer-review process: an actor LLM first generates an initial response to a query, and critic peer LLMs evaluate and provide feedback on the response. The actor revises or corrects its response based on this feedback, and the critics finally assign scores to the revised response. The scores of the initial and revised outputs are used as preference scores to construct preference data. This process enables autonomous and collective reasoning of LLMs for constructing preference data without human supervision. However, preference data constructed by LLMs may be subject to noise or reward hacking. To mitigate the issue, we first provide a unified view on robust preference learning through the lens of risk minimization, and then propose a framework for robust training on self-correction datasets. Experiments show that PULSE significantly outperforms existing approaches, achieving performance gains up to 47.3% and 34.6% on Alpaca LC and Alpaca 2.0, and 23.9%, 102.8%, and 12.4% on a collection of math, coding, and general reasoning tasks, demonstrating its potential to create and sustain scalable LLM ecosystems.

## 1 INTRODUCTION

Large language models (LLMs) have become a foundational tool for modern AI, enabling cutting-edge capabilities in various real-world applications, including chatbots, coding assistants, and reasoning systems Anthropic (2024); OpenAI (2023); Team et al. (2023). As such, it is important to align LLMs with human values and preferences. Reinforcement Learning from Human Feedback (RLHF) Liu et al. (2020); Ouyang et al. (2022) is a key method for preference alignment based on human feedback. Subsequently, Direct Preference Optimization (DPO) Rafailov et al. (2023) was proposed, which directly optimizes LLMs for human preferences without RL and has been widely adopted as a simple yet efficient alternative to RL-based methods. However, alignment techniques rely on high-quality, human-annotated preference datasets, which pose a significant challenge for training LLMs at scale.

To this end, various *self-improvement* methods have been proposed for LLMs. A common approach involves generating synthetic datasets and fine-tuning the model on them. Annotating those datasets typically relies on either LLMs serving as judges Bai et al. (2022); Yuan et al. (2024) or supervised reward models Snorkel (2024) aligned with human feedback; both of which incur substantial resource costs and limit scalability. Moreover, these works tend to overlook the correction of original responses. Zelikman et al. (2022); Kumar et al. (2025) proposed learning methods based on *self-correction*. They iteratively refine their generated responses from corrections without relying on human supervision, and train the models on refined, high-quality data from both correct and incorrect generations. These approaches require ground-truth answers, which may not always be available. Lack of answers is common for questions in open-ended or underexplored domains. However, such questions can be a valuable resource for the sustainable improvement of AI systems. In this context, multi-agent LLM systems Bo et al. (2024); Gao et al. (2023a) can collaboratively discuss such questions, offering feedback, challenging assumptions, and iteratively refining their responses. We envision that this form of autonomous and collective reasoning can advance problem-solving and generalization capabilities.

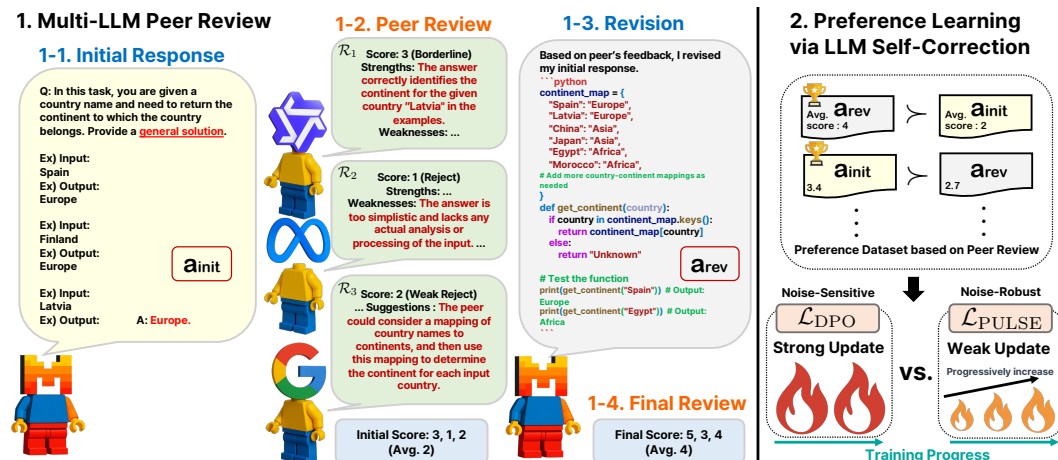

Figure 1: **Overview of PULSE.** During the LLM peer review process: **1)** the actor LLM generates an initial response to the question; **2)** the critic LLMs review the response and request a revision; **3)** the actor LLM revises its initial response; **4)** the critic LLMs assign a final score and construct the preference dataset; **5)** The LLM is updated with proposed loss $\mathcal{L}_{\text{PULSE}}$ at a slow and gradual pace for a lower sensitivity to noisy data compared to DPO. $a \succ b$ means response $a$ is preferred to response $b$.

**Contribution.** We propose **P**eer-g**U**ided Preference **L**earning via LLM **SE**lf-correction (**PULSE**), a novel self-improvement framework based on the academic peer-review process, enabling LLMs to refine themselves through collaborative review and preference optimization. In PULSE, an actor LLM first generates an initial response, which is then reviewed by multiple peer LLMs. The actor revises its output based on the peers' feedback. Finally, critic LLMs evaluate the quality of the revised response and produce a preference signal that serves as training supervision. This pipeline enables the construction of high-quality preference datasets without relying on external ground-truth answers or costly human annotations. Furthermore, PULSE supports both success and failure cases, encouraging the model to not only reinforce correct reasoning but also learn from its mistakes. However, there exists a practical issue with preference learning on datasets constructed by LLMs: preference judgments by LLMs may be subject to noise Wu et al. (2025) or length bias Meng et al. (2024). To address this issue, we propose a framework for robust preference learning leveraging the theory of *risk minimization* for classification tasks Zhang (2004). As a result, PULSE adopts a simple DPO-style training offering a scalable approach to preference alignment for self-improvement with built-in robustness. Experiments demonstrate that PULSE significantly outperforms existing approaches across comprehensive instruction-following benchmark datasets. Specifically, PULSE achieves performance gains up to 72.7% over base LLMs, and up to 47.3% and 34.6% on Alpaca LC and Alpaca 2.0 with Mistral-7B over previous self-training methods that rely on external reward models. Importantly, PULSE achieves the performance improvement compared to the runner-up method up to 23.9%, 102.8%, and 12.4% on math, code, and general reasoning tasks.

## 2 PROPOSED METHOD

**Motivation.** As LLMs continue to advance, traditional learning paradigms that rely on human supervision face the *supervision bottleneck* caused by the poor scalability of human annotation. To overcome this, we propose to shift the focus from traditional learning (e.g., human-to-model Ouyang et al. (2022)) toward a collaborative paradigm in which **peer LLMs of similar capacity supervise and refine each other**. This *peer-to-peer learning* strategy offers a scalable alternative to human-guided learning. We envision a system where intelligent agents co-evolve through interaction, negotiation, and reflection, similar to human societies.

### 2.1 PREFERENCE DATA WITH PEER-GUIDED SELF-CORRECTION

**Overview.** We introduce **PULSE: P**eer-g**U**ided Preference **L**earning via LLM **SE**lf-Correction, a novel framework inspired by the academic peer-review process. The goal is to construct high-quality

preference data with an actor LLM and to optimize the LLM via preference learning on the self-generated data. The actor LLM is guided by the critiques of peer LLMs as follows. For a group of participant LLMs, one LLM becomes an actor LLM, and the others become critic LLMs. The actor LLM generates responses to questions, and the critic LLMs act as independent reviewers, evaluating and providing constructive feedback on the actor's responses. The reviews encompass explicit strengths, weaknesses, and actionable suggestions. A summary of the process is as follows.

1. **Initial Response and Peer Review:** The actor LLM first generates an initial response to the given question. The peer-critic LLMs evaluate the response, assigning scores on a 5-point ordinal scale: 5 (accept), 4 (weak accept), 3 (borderline), 2 (weak reject), and 1 (reject). In addition to scores, each reviewer provides structured feedback detailing the responses' strengths, weaknesses, and proposed revisions.

2. **Self-Correction and Final Evaluation:** The actor LLM incorporates the reviewers' feedback to produce a revised (self-corrected) response. The same set of peer reviewers then reassess the revised output, offering updated scores and rationales to judge the degree of improvement over the initial response.

3. **Preference Dataset Construction and Fine-Tuning:** After completing this two-stage evaluation, a preference dataset is constructed. Each sample consists of a question, the initial response, the revised response, and their preferences. Between the initial and revised responses, the one with the higher score is annotated as the preferred response. Then the actor LLM is fine-tuned on the dataset through preference optimization.

This peer-guided mechanism enables iterative refinement of the actor's outputs without reliance on supervised datasets or human annotations. Fig.1 depicts an overview of PULSE.

**Detailed Review Process.** We describe the detailed review process. Suppose the LLMs are indexed by $k \in [K]$. Let $\pi_\theta^k$ denote the actor LLM, and $\pi_{\text{peer}}^j$ for $j \in [K] \setminus \{k\}$ denote the critic LLMs. $\mathcal{P}$ denotes a prompt template, and $\mathcal{M}$ denotes the response to an input prompt.

**Step 1: Initial Response Generation.** Given an input question $q$, the actor LLM $\pi_\theta^k$ first generates an initial response $a_{\text{init}}$ using a prompt template $\mathcal{P}_{\text{answer}}$:

$$a_{\text{init}} = \mathcal{M}(\mathcal{P}_{\text{answer}}(q, \pi_\theta^k)) \tag{1}$$

**Step 2: Independent Peer Review.** Each critic LLM $\pi_{\text{peer}}^j$ independently evaluates the initial response $a_{\text{init}}$ by assigning a numerical score $s_j(a_{\text{init}}) \in \{1, 2, 3, 4, 5\}$ and composing a textual review $\mathcal{R}_j$ that highlights its strengths, weaknesses, and recommended suggestions. Importantly, each reviewer only observes $a_{\text{init}}$, not the other reviews, ensuring that evaluations remain independent and unbiased:

$$\mathcal{R}_j, s_j(a_{\text{init}}) = \mathcal{M}(\mathcal{P}_{\text{review}}(a_{\text{init}}, \pi_{\text{peer}}^j)), \quad j \in [K] \setminus \{k\} \tag{2}$$

**Step 3: Actor Self-Revision.** The actor LLM $\pi_\theta$ then synthesizes the set of all reviews into a single concatenated meta-review $\hat{\mathcal{R}}^*$ and generates a revised response $a_{\text{rev}}$ that aims to address the feedback and improve upon $a_{\text{init}}$:

$$a_{\text{rev}} = \mathcal{M}(\mathcal{P}_{\text{revise}}(q, a_{\text{init}}, \hat{\mathcal{R}}^*, \pi_\theta^k)) \tag{3}$$

This process resembles the reflective refinement loop proposed in interactive agent systems Bo et al. (2024); Shinn et al. (2023), but without human intervention.

**Step 4: Final Evaluation.** To assess the effectiveness of the revision, each reviewer independently re-evaluates the revised response $a_{\text{rev}}$, again without access to other reviews or prior evaluations. Each review includes both a score $s_j(a_{\text{rev}})$ and a rationale $\hat{\mathcal{R}}_j$ that explains the grading decision:

$$\hat{\mathcal{R}}_j, s_j(a_{\text{rev}}) = \mathcal{M}(\mathcal{P}_{\text{review}}(a_{\text{rev}}, \pi_{\text{peer}}^j)), \quad j \in [K] \setminus \{k\} \tag{4}$$

The purpose of requesting critic LLMs to generate rationale $\hat{\mathcal{R}}_j$ is to guide the critic LLMs to produce appropriate scores based on step-by-step reasoning, similar to CoT prompting Wei et al. (2022).

**Step 5: Preference Dataset Construction.** We compute the average score $s_*(\cdot)$ across reviewers to determine which of the initial and revised responses is preferred. Let us denote the winning (preferred) response by $a_w$ and the losing (dispreferred) response by $a_l$.

$$(a_w, a_l) = \begin{cases} (a_{\text{rev}}, a_{\text{init}}) & \text{if } s_*(a_{\text{rev}}) > s_*(a_{\text{init}}) \\ (a_{\text{init}}, a_{\text{rev}}) & \text{otherwise} \end{cases} \tag{5}$$

The prompt templates used for instruction and review stages are provided in Appendix A.11. Detailed examples of review and revision interactions are provided in Appendix A.15.

## 2.2 A Unified View to Preference Learning as Robust Binary Classification

**Challenges in Preference Learning.** Multi-agent approaches, such as PULSE, are crucial for the scalability of LLM preference learning. On the other hand, preference data constructed by LLMs may have the following issues.

- **Noise in Preference Judgment:** In preference learning, the criteria for preference judgment may vary across evaluators in the absence of a predefined agreement. This may result in inconsistent or noisy labels, when using AI or even human feedback Guo et al. (2024); Wu et al. (2024).

- **Reward Hacking from Length Bias:** The revised response ($a_{\text{rev}}$) is typically longer than the initial response ($a_{\text{init}}$), because it adds new information while correcting the original flaws. This may create a signal where "longer is better." This bias can cause the model to perform *reward hacking* which exploits the length bias to maximize its reward without improving quality Skalse et al. (2022); Gao et al. (2023b); Meng et al. (2024).

For example, an inspection of our preference dataset showed that the revised responses are 38.9% longer than the initial responses. These observations suggest that preference datasets may be subject to noisy signals. In this section, we propose a robust preference learning for PULSE.

**Preliminaries.** Direct Preference Optimization (DPO) Rafailov et al. (2023) provides a scalable and principled alternative to RLHF Ouyang et al. (2022), avoiding the need for RL or explicit reward modeling. Let $\pi_\theta$ denote a trainable actor LLM and $\pi_{\text{ref}}$ a fixed reference model. Suppose $q$ is a prompt, $a_w$ is the preferred (winning) response, and $a_l$ is the dispreferred (losing) response (denoted as preference relation $a_w \succ a_l | q$). DPO optimizes the following objective:

$$\mathcal{L}_{\text{DPO}}(q, a_w, a_l; \theta) = -\log \left[ \frac{1}{1 + \exp\left[ -\left( \beta \log \frac{\pi_\theta(a_w|q)}{\pi_{\text{ref}}(a_w|q)} - \beta \log \frac{\pi_\theta(a_l|q)}{\pi_{\text{ref}}(a_l|q)} \right) \right]} \right] \tag{6}$$

where $\beta > 0$ is a hyperparameter controlling the sharpness of preference. In the following, we discuss a framework for robust preference learning, and also show that DPO may impose a substantial penalty on mislabeled samples, making the preference learning susceptible to noisy data.

**Risk Minimization.** Consider the task of deciding the preferred answer between $a_{\text{rev}}$ and $a_{\text{init}}$ given prompt $q$. We frame our task as risk minimization for binary classification following the framework in Masnadi-Shirazi & Vasconcelos (2008); Zhang (2004). Consider the classification problem given input $x \in \mathcal{X}$ where $\mathcal{X}$ is a space of tuples $(q, a_{\text{rev}}, a_{\text{init}})$ and label $y \in \{+1, -1\}$ defined as

$$y = \begin{cases} +1, & a_{\text{rev}} \succ a_{\text{init}} \mid q \\ -1, & \text{otherwise} \end{cases} \tag{7}$$

where preference relation $a_{\text{rev}} \succ a_{\text{init}}$ is based on review scores, i.e., $s_*(a_{\text{rev}}) > s_*(a_{\text{rev}})$. A binary classifier is designed with a discriminant function $f : \mathcal{X} \to \mathbb{R}$ which maps inputs to a real number. The goal is to choose $f(x)$ whose sign can accurately predict the label, i.e., $f(x) \geq 0$ if and only if $y = +1$. The classifier can be trained by minimizing the *risk* given by $\mathbb{E}_{X,Y}[\phi(Yf(X))]$ with loss function $\phi$. The loss measures the disagreement between the signs of $f(x)$ and $y$.

The risk minimization is equivalent to minimizing the *conditional risk* as follows. Let $\eta(x) := P(Y = +1|x)$ denote the posterior probability.

$$\mathbb{E}_{Y|x}[\phi(Yf(x))] = \eta \cdot \phi(f) + (1 - \eta) \cdot \phi(-f) \tag{8}$$

given feasible input $x$ (the dependence of $\eta$ and $f$ on $x$ is omitted for brevity). Let $C$ denote the minimum conditional risk achievable over discriminant functions:

$$C(\eta) := \inf_f \left[ \mathbb{E}_{Y|x}[\phi(Yf(x))] \right] = \inf_f \left[ \eta \cdot \phi(f) + (1 - \eta) \cdot \phi(-f) \right] \tag{9}$$

$C$ is a concave function irrespective of $\phi$, because it is an infimum of concave functions in $\eta$. While $C$ is determined from $\phi$ through (9), the process can be "reversed", i.e., loss $\phi$ can be recovered from $C$. This remarkable result was provided by Masnadi-Shirazi & Vasconcelos (2008) and is given by

$$\phi(f) = C(p(f)) + (1 - p(f)) \cdot C'(p(f)) \tag{10}$$

where $p(f) \in [0, 1]$ is an estimate of the posterior given discriminant $f$. Importantly, one can "design" loss $\phi$ with some desirable properties by properly selecting risk function $C$ and posterior model $p(f)$. A theoretical implication is that, the risk minimization with loss $\phi(f)$ obtained by (10) is equivalent to minimizing the distance between the estimate $p(f)$ and the true posterior $\eta$, where the distance is measured by Bregman divergence with respect to $-C$ Masnadi-Shirazi & Vasconcelos (2008); Zhang (2004). The choices of $C$ and $p(f)$ have been studied for SVM, boosting, logistic regression, etc.: we refer the readers to Zhang (2004); Masnadi-Shirazi & Vasconcelos (2008) for details.

**Loss function for PULSE.** The key task is to choose (i) a discriminant function $f$, (ii) a posterior model $p(f)$, (iii) minimum conditional risk $C$, so as to find loss $\phi$ using (10) for robust preference learning. We begin with a latent reward model, i.e., there exists some latent reward $r$ modeling human preferences. Denote the latent reward of answer $a$ given question $q$ by $r(q, a)$. Since the decision is based on the sign of $f(x)$, we propose to define discriminant function $f(x)$ as

$$f(x) = r(q, a_{\text{rev}}) - r(q, a_{\text{init}}), \quad \text{for } x = (q, a_{\text{rev}}, a_{\text{init}}) \tag{11}$$

where one decides on $a_{\text{rev}}$ or $a_{\text{init}}$ which returns the higher latent reward. Next, we define the posterior estimate $p(f)$. Let $\sigma(\cdot)$ denote the sigmoid function. Our choice of $p(f)$ is

$$p(f) = \sigma(f) = \frac{1}{1 + e^{-f}} = \frac{\exp(r(q, a_{\text{rev}}))}{\exp(r(q, a_{\text{rev}})) + \exp(r(q, a_{\text{init}}))} \tag{12}$$

to which we apply (11) for the last equality. Note that (12) is equivalent to $P(a_{\text{rev}} \succ a_{\text{init}}|q)$ under Bradley-Terry (BT) model Bradley & Terry (1952), which is widely used for preference modeling, e.g., DPO. Finally, we choose the minimum conditional risk $C$ as a quadratic function given by

$$C(p) = p(1 - p) \tag{13}$$

where quadratic risk $C$ is associated with a squared loss Bartlett et al. (2006); Zhang (2004). Indeed, if we apply $C(p)$ to (10), we obtain a loss $\phi$ quadratic in posterior estimate $p(f)$ as follows.

$$\phi_{\text{quad}}(f) = (1 - p(f))^2 = (1 - \sigma(f))^2 = \frac{1}{(1 + e^f)^2} \tag{14}$$

If we apply (14) to (8), the conditional risk is given by

$$\eta(1 - \sigma(f))^2 + (1 - \eta)(1 - \sigma(-f))^2 = \mathbb{E}_{Y|x}[\{\mathbf{1}(Y = +1) - \sigma(f(x))\}^2] \tag{15}$$

where $\mathbf{1}(A)$ is the 0-1 indicator of condition $A$. (15) represents the Mean Squared Error (MSE) of posterior $p(f)$ and is also called the Brier score Brier (1950). Next, we use the reward model proposed by DPO, which captures the regularization of $\pi_\theta$ by reference model $\pi_{\text{ref}}$ given by

$$r(q, a) = \beta \left[\log \pi_\theta(a|q) - \log \pi_{\text{ref}}(a|q)\right] + \beta \log Z(q) \tag{16}$$

where $Z(q)$ denotes the partition function Rafailov et al. (2023). We obtain the final loss by applying (11), (14), and (16) to $\phi_{\text{quad}}(y \cdot f(x))$. Denote the preferred answer between $a_{\text{rev}}$ and $a_{\text{init}}$ by $a_w$, and the dispreferred by $a_l$. The loss $\phi_{\text{quad}}(y \cdot f(x))$ for $x = (q, a_{\text{rev}}, a_{\text{init}})$, denoted by $\mathcal{L}_{\text{PULSE}}(x, y; \theta)$, is

$$\mathcal{L}_{\text{PULSE}}(x, y; \theta) = \frac{1}{\left[1 + \exp\left(\beta \log \frac{\pi_\theta(a_w|q)}{\pi_{\text{ref}}(a_w|q)} - \beta \log \frac{\pi_\theta(a_l|q)}{\pi_{\text{ref}}(a_l|q)}\right)\right]^2} \tag{17}$$

Next, we explain the robustness associated with PULSE.

**Robustness of PULSE: Boundedness and Concavity.** We first formulate DPO as a risk minimization. Suppose we choose discriminant $f$ as (11), the posterior as (12), but $C(p)$ as the binary entropy or $C(p) = -p \log p - (1 - p) \log(1 - p)$. This yields $\phi$ logarithmic in $p$:

$$\phi_{\text{log}}(f) = -\log p(f) = -\log\left(\frac{1}{1 + e^{-f}}\right) \tag{18}$$

If we apply (11) and (16) to (18), we obtain the DPO loss (6) from $\phi_{\text{log}}(y \cdot f(x))$. Thus, DPO can be viewed as a risk minimization using the difference in latent rewards as discriminant $f$, $\sigma(f)$ as the posterior estimate, and the binary entropy as the risk function. The conditional risk is given by

$$\mathbb{E}_{Y|x}[-\mathbf{1}(Y = +1) \log(\sigma(f(x)) - \mathbf{1}(Y = -1) \log(1 - \sigma(f(x)))]$$

Thus, the risk minimization in DPO is equivalent to the maximum likelihood estimation of the posterior under the BT model.

The key difference between PULSE and DPO is the choice of $C$ and resulting losses: $\phi_{\text{quad}}$ in (14) for PULSE and $\phi_{\text{log}}$ in (18) for DPO. We focus on the high-risk regime regarding robustness; since the risk is $\phi(y \cdot f(x))$, the risk of a "severe" misclassification is $\phi(f)$ with $f$ being a negative number of large magnitude. As $f$ tends to $-\infty$, $\phi_{\text{log}}$ for DPO is: **1)** approximately linear in $f$ and thus is unbounded, and **2)** its gradient increases because $\phi_{\text{log}}$ is convex in $f$. In contrast, $\phi_{\text{quad}}$ for PULSE is: **1)** bounded above by 1, and **2)** its gradient decreases because $\phi_{\text{quad}}$ is concave for $f < -\log 2$ (see Fig. 2). Thus, in case of misclassification during training, DPO aggressively

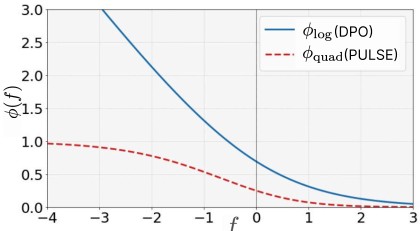

Figure 2: Comparison of $\phi_{\text{quad}}$ (PULSE) and $\phi_{\text{log}}$ (DPO).

penalizes the model, whereas PULSE is less sensitive to large errors. Since errors can be induced due to various noise in the self/peer-generated data, it is preferable to be less sensitive to error signals for robust training. Similar observations on robustness are made for Savage loss Masnadi-Shirazi & Vasconcelos (2008) which combines a squared risk and boosting, while our loss represents the MSE of the sigmoidal posterior from the BT preference model.

**Robustness of PULSE: $\beta$-Scheduling.** We propose the scheduling of hyperparameter $\beta$ in (17). The larger $\beta$ means that the loss has a higher sensitivity to latent rewards, and thus to errors as well. Thus, we will make $\beta$ small in the early training phase when the model's confidence in preference is low, and we gradually restore $\beta$ to a target value. This makes the model update less sensitive to noisy data. We propose a *linear scheduling* of $\beta$. Specifically, we replace $\beta$ in (17) by $\beta_\tau$ given by

$$\beta_\tau = \beta \cdot \min(1, \delta\tau) \quad \tau = 1, 2, 3, ... \tag{19}$$

where $\tau$ is the number of training steps and $\delta > 0$ is the slope. Thus, $\beta_\tau$ increases linearly over the training steps to the target value $\beta$.

Another justification for $\beta$-scheduling in connection with risk minimization is as follows. A robust classification under noisy labels is studied in Ghosh et al. (2015) which showed that risk minimization with loss $\Phi(x, y)$ for input $x$ and label $y \in \{+1, -1\}$ is *noise-tolerant* under uniform noise[1] if

$$\Phi(x, -y) + \Phi(x, y) = K \tag{20}$$

Ghosh et al. (2015) also showed that $\Phi$ with condition (20) is approximately noise-tolerant under non-uniform noise as well, if the minimum achievable risk (upper-bounded by the converged loss) is small. (20) implies that $\Phi$ should be bounded, which also holds for our loss $\mathcal{L}_{\text{PULSE}}$. Moreover, we show that $\mathcal{L}_{\text{PULSE}}$ satisfies (20) for small $\beta$ up to its first order (a derivation is provided in Appendix A.1):

$$\mathcal{L}_{\text{PULSE}}(x, y; \theta) + \mathcal{L}_{\text{PULSE}}(x, -y; \theta) = \frac{1}{2} + o(\beta) \tag{21}$$

Thus, our loss approximately satisfies (20) in the early stage of training (small $\beta$), during which robust training is crucial. In practice, fixing $\beta$ to a small value would make training slow. Thus, we gradually increase $\beta$ as the model becomes more aligned with preferences through $\beta$-scheduling.

Overall, we observe that the loss of PULSE leads to more stable training than DPO. An example plot of the gradient norm during training is provided in Fig. 7 in Appendix A.5, which shows that PULSE performs more stable updates to the model than DPO: see Appendix A.5 for more details.

**PULSE improves the training on peer-review data.** The proposed peer-review data enjoy high quality self-corrections that improve reasoning performance. However, the revision process by self-correction naturally produces longer responses, which can introduce a mild length bias. As shown in the above, the proposed training mitigates this issue by making the early stage of preference learning robust to such spurious signals, allowing the model to fully exploit the benefits of diversified self-corrections while avoiding length bias.

**Additional study on loss with logarithmic rates.** We argued that the robustness lies in the rate at which $\phi$ increases as $f \to -\infty$: $\phi(f) \sim 1$ (constant) for PULSE and $\phi(f) \sim |f|$ (linear) for DPO. A

---

[1]A risk minimization is noise-tolerant if the classifier minimizing the risk has the same probability of error with or without noise in the labels. The noise is *uniform* if the probability of the label being incorrect is the same for all data samples, and *non-uniform* if it is sample-dependent. See Ghosh et al. (2015) for details.

natural question is: Are there $\phi$'s with different rates? We examined a loss that scales as $\sim \log |f|$ as $f \to -\infty$, making its sensitivity to errors between PULSE and DPO. The analysis of the loss from the perspective of risk minimization and the related discussion are provided in Appendix A.2.

**Comparison with noise-robust methods.** Recently, a line of work was proposed for noise-robustness in preference learning, e.g., rDPO Chowdhury et al. (2024), PerpCorrect Kong et al. (2024), cDPO Mitchell (2023). The methods rely on the key assumption that labels are flipped at random with a fixed rate $\epsilon$ and on the estimation of $\epsilon$. This is somewhat strong assumption in practice; in real-world datasets, the probability of error is likely to depend on input data. In contrast, PULSE is not constrained by $\epsilon$-assumption nor does it need to estimate $\epsilon$ to achieve noise robustness. More detailed discussions on these methods are provided in Appendix A.8.

## 3 EXPERIMENT

### 3.1 EXPERIMENTAL SETTINGS

**Models.** We experiment with four instruction-tuned models: Mistral-7B-v0.2 Jiang et al. (2023a), LLaMA3-8B Grattafiori et al. (2024), Qwen2.5-7B Yang et al. (2024), and Gemma2-9B Team et al. (2024) as the initial model $\pi_\theta$ and reviewer models $\pi_{\text{peer}}^k$. When one model acts as an actor LLM, the other three models act as critic/reviewer LLMs. We use the same hyperparameters for all four LLMs, as listed in Appendix A.12.

**Baselines.** We compare SoTA preference reward models such as Snorkel Snorkel (2024) and Skywork-Reward Liu et al. (2025). Comparisons cover models trained with Iterative DPO, IPO Snorkel (2024); Xu et al. (2023). To assess the effectiveness of our peer review-based preference dataset construction and learning method, we compare PULSE with DPO Rafailov et al. (2023) using identical datasets. For Iterative DPO and IPO, we perform a three-round iterative training process using the curated UltraFeedback dataset Cui et al. (2024). In each iteration, we generate 5 responses with a temperature of 1.0 and $p = 1.0$ and select the preferred and rejected responses based on the highest and lowest scores assigned by PairRM Snorkel (2024).

**Datasets.** For generating a preference learning dataset through the peer review process, we adopted the prompts from UltraFeedback Cui et al. (2024). From the generated dataset, we discard samples such that the preferred response received a score below a predefined threshold (e.g., 3 out of 5) to ensure high quality. This filtering excluded at most 1.2% of all examples. *We did not use any ground-truth responses* to assess the effectiveness of the self-correction mechanism induced solely by the peer review process.

**Evaluations.** We used Alpaca LC Dubois et al. (2024) to reduce length bias in preference judgments Wang et al. (2023); Zheng et al. (2023). This metric normalizes the effects of the response length via a regression model, isolating response quality. We also report results on AlpacaEval 2.0 Dubois et al. (2023); Li et al. (2023b) for comparison with existing baselines. Both Alpaca LC and AlpacaEval 2.0 estimate human preferences by comparing model outputs with GPT-4 responses. Furthermore, we evaluated with MT-Bench Zheng et al. (2023) to measure broader capabilities such as math, coding, role-play, and writing. We followed standard evaluation protocol shown in the Appendix A.14. Finally, we evaluated methods on diverse reasoning tasks, such as math: GSM8K Cobbe et al. (2021), MATH Lewkowycz et al. (2022), coding: HumanEval Chan et al. (2024), MBPP Austin et al. (2021), general reasoning: GPQA Rein et al. (2024), and MMLU Hendrycks et al. (2020). (See Appendix A.13 for shots).

**Main Results.** Table 1 shows that PULSE exceeds the performance of PairRM-based tuning by up to 47.3%, 34.6%, and 9.6% on Alpaca LC, Alpaca 2.0, and MT-Bench, respectively. Also, PULSE outperforms DPO trained by Skywork-Reward by 7.6%, 6.8%, and 2.9% on the same datasets. We attribute the strong performance of PULSE to the quality of the peer-review dataset, which was constructed from multiple LLMs to enhance diversity and reduce sampling bias. Table 2 shows that PULSE achieves the best performance across diverse reasoning tasks. PULSE significantly outperforms Skywork-DPO, which is the second-best model in our experiments, on math tasks: GSM8K on 5.5%, MATH on 23.9%, code tasks: HumanEval on 102.8%, MBPP on 96.5%, and reasoning tasks: GPQA on 12.4%, MMLU on 5.3%. This result underscores the effectiveness of a collaborative self-correction in improving the reasoning abilities of LLMs.

**Effectiveness of Peer-Review.** Next, we examine the effectiveness of our preference data constructed by self-correction guided by peer-review. Table 3 shows that our peer review process consistently

Table 1: Main results of Alpaca LC, 2.0, and MT Bench. All methods are trained using Mistral-7B-Instruct-v0.2.

| Methods | Pref. judgment | Loss | Alpaca LC. | Alpaca 2.0. | MT Bench |
|---|---|---|---|---|---|
| Mistral-7B | - | - | 17.11 | 14.72 | 7.51 |
| Iterative-DPO | PairRM | DPO | 20.06 | 22.47 | 7.09 |
| Iterative-IPO | PairRM | IPO | 22.30 | 23.39 | 7.14 |
| Snorkel | PairRM | DPO | 26.29 | 29.82 | 7.58 |
| Skywork-DPO | Skywork-Reward | DPO | 27.45 | 28.32 | 7.55 |
| PeerReview-DPO | PeerReview | DPO | 23.75 | 28.38 | 7.63 |
| **PULSE** | PeerReview | PULSE | **29.54** | **30.24** | **7.77** |

Table 2: Main results of diverse math, code, and reasoning tasks. All methods are trained using Mistral-7B-Instruct-v0.2.

| Methods | Pref. judgment | Loss | GSM8K | MATH | HumanEval | MBPP | GPQA-Diamond | MMLU |
|---|---|---|---|---|---|---|---|---|
| Mistral-7B | - | - | 44.50 | 10.02 | 10.98 | 14.20 | 23.74 | 55.96 |
| Iterative-DPO | PairRM | DPO | 36.43 | 7.27 | 2.75 | 15.80 | 26.72 | 52.00 |
| Iterative-IPO | PairRM | IPO | 41.12 | 9.45 | 3.53 | 16.27 | 27.54 | 53.57 |
| Snorkel | PairRM | DPO | 40.03 | 8.32 | 4.88 | 17.00 | 27.78 | 54.77 |
| Skywork-DPO | Skywork-Reward | DPO | 45.13 | 10.25 | 8.72 | 19.24 | 28.32 | 56.62 |
| PeerReview-DPO | PeerReview | DPO | 45.09 | 10.16 | 9.10 | 32.80 | 27.82 | 56.16 |
| **PULSE** | PeerReview | PULSE | **47.61** | **12.70** | **17.68** | **37.80** | **31.82** | **57.53** |

Table 3: **Left.** Main results of Alpaca LC and the corresponding average response lengths. Blue indicates the performance improvement over the base LLMs, while Red shows the increase in average response lengths compared to the base LLMs. **Right.** Main results of MT-Bench. In both tables, **bold** denotes the best performance, and underline indicates the second-best performance.

| Model | Loss | Alpaca LC. | Avg. len |
|---|---|---|---|
| Mistral-7B | - | 17.11 | 1676 |
| Mistral-7B (PeerReview) | DPO | 23.75 (+6.64) | 2540 (+864) |
| Mistral-7B (PeerReview) | PULSE | **29.54** (+12.43) | 2096 (+420) |
| LLaMA3-8B | - | 22.32 | 1899 |
| LLaMA3-8B (PeerReview) | DPO | 23.78 (+1.46) | 2926 (+1027) |
| LLaMA3-8B (PeerReview) | PULSE | **27.39** (+5.07) | 2254 (+355) |
| Qwen2.5-7B | - | 27.42 | 1943 |
| Qwen2.5-7B (PeerReview) | DPO | 31.15 (+3.73) | 2459 (+516) |
| Qwen2.5-7B (PeerReview) | PULSE | **33.84** (+6.42) | 2216 (+273) |
| Gemma2-9B | - | 45.08 | 1590 |
| Gemma2-9B (PeerReview) | DPO | 51.14 (+6.06) | 2249 (+659) |
| Gemma2-9B (PeerReview) | PULSE | **53.85** (+8.77) | 2037 (+447) |

| Model | Loss | 1st turn | 2nd turn | Avg. |
|---|---|---|---|---|
| Mistral-7B | - | 7.78 | 7.25 | 7.51 |
| Mistral-7B (PeerReview) | DPO | 7.74 | 7.52 | 7.63 |
| Mistral-7B (PeerReview) | PULSE | **7.92** | **7.62** | **7.77** |
| LLaMA3-8B | - | 8.36 | 7.76 | 8.06 |
| LLaMA3-8B (PeerReview) | DPO | 8.53 | 7.85 | 8.19 |
| LLaMA3-8B (PeerReview) | PULSE | **8.64** | **7.88** | **8.26** |
| Qwen2.5-7B | - | 8.80 | 8.34 | 8.57 |
| Qwen2.5-7B (PeerReview) | DPO | 9.02 | 8.45 | 8.74 |
| Qwen2.5-7B (PeerReview) | PULSE | **9.05** | **8.53** | **8.79** |
| Gemma2-9B | - | 8.81 | 8.23 | 8.52 |
| Gemma2-9B (PeerReview) | DPO | 9.01 | 8.56 | 8.79 |
| Gemma2-9B (PeerReview) | PULSE | **9.02** | **8.60** | **8.81** |

improves all four LLMs under both DPO and PULSE, yielding performance gains of up to 72.7% on Mistral-7B, 22.7% on LLaMA3-8B, 23.4% on Qwen2.5-7B, and 19.5% on Gemma2-9B for Alpaca LC. Similarly, Table 3 shows improvements of up to 3.4% on MT-Bench. Results indicate that PULSE can enhance not only a specific model but also diverse LLMs through collaboration. Additionally, we compare two training methods: PULSE and DPO. The results show that PULSE consistently outperforms peer-guided DPO, achieving up to 24.3% higher on Alpaca LC and 2% on MT-Bench. Notably, DPO tends to be unnecessarily verbose; it produces much longer outputs, between 126.6%-154.1% longer than the base models. In contrast, PULSE produces more concise responses (118.7-128.1%). The result demonstrates that PULSE effectively mitigates reward hacking from length bias, improving training stability and facilitating better generalization.

**Ablation study.** We conducted an ablation study on the two main components of PULSE: the peer-review dataset and $\beta$-scheduling (Table 4). A variant of PULSE trained with only the peer-review component achieved significant gains of up to 66.9% on Alpaca LC and 3.4% on MT-Bench, demonstrating the effectiveness of our dataset. Further incorporating $\beta$-scheduling yielded additional improvements of up to 4.2% and 2.0% respectively, which suggests that progressively strengthening

Table 4: **Ablation study.** PeerReview refers to the peer review process described in Sec.2.1, and $\beta_\tau$ denotes the $\beta$-scheduling in Eq.19 (see Appendix A.12 for the settings of $\beta$ and $\delta$ for PULSE.)

| PeerReview | $\beta_\tau$ | Mistral-7B | | LLaMA3-8B | | Qwen2.5-7B | | Gemma2-9B | |
|---|---|---|---|---|---|---|---|---|---|
| - | - | Alpaca LC. | MT Bench | Alpaca LC. | MT Bench | Alpaca LC. | MT Bench | Alpaca LC. | MT Bench |
| ✗ | ✗ | 17.11 | 7.51 | 22.32 | 8.06 | 27.42 | 8.57 | 45.08 | 8.52 |
| ✔ | ✗ | 28.55 | 7.62 | 26.57 | 8.15 | 32.47 | 8.64 | 53.24 | 8.75 |
| ✔ | ✔ | **29.54** | **7.77** | **27.39** | **8.26** | **33.84** | **8.79** | **53.85** | **8.81** |

| Method | Alpaca LC. | GSM8K | MATH | HumanEval | MBPP | MMLU |
|---|---|---|---|---|---|---|
| PULSE w/ self-critic | 19.25 | 44.27 | 9.65 | 10.27 | 25.40 | 53.28 |
| PULSE w/ single critic | 21.17 | 44.68 | 11.10 | 15.07 | 30.60 | 55.63 |
| PULSE | **29.54** | **47.61** | **12.70** | **17.68** | **37.80** | **57.53** |

Table 5: Ablation study on the diversity of critics. Mistral-7B is the actor LLM. "Self-critic" means that the actor takes the critic role as well. "Single-critic" means that there is only one reviewer (LLaMA3-8B).

| Method | Alpaca LC. | Alpaca 2.0 | GSM8K | MATH | MBPP | MMLU |
|---|---|---|---|---|---|---|
| Mistral-7B | 17.11 | 14.72 | 44.50 | 10.98 | 14.20 | 55.96 |
| PULSE w/ Ultrafeedback | 21.49 | 19.67 | 44.65 | 11.56 | 34.03 | 55.42 |
| PULSE w/ HH-RLHF | 23.61 | 26.59 | 46.07 | 11.84 | 32.00 | 55.61 |
| PULSE w/ PeerReview | **29.54** | **30.24** | **47.61** | **12.70** | **37.80** | **57.53** |

Table 6: Effect of applying the PULSE loss to existing preference datasets.

the learning signal helps mitigate overfitting to noisy data. All results confirm that both components are crucial contributors to PULSE's overall performance.

**Ablation study of critic models.** We further investigate the impact of the multi-agent setup in PeerReview datasets in Table 5. The results show that the multi-agent configuration is crucial: both self-critic (actor is also critic) and single-critic (one reviewer) variants achieve noticeably lower performance compared to the default multi-agentic approach.

**PULSE with other preference datasets.** We evaluate the robustness of loss $\mathcal{L}_{\text{PULSE}}$ applied to existing preference datasets. Through this evaluation, we can assess the robustness of PULSE not only in peer-reviewed but also in conventional preference datasets. We trained Mistral-7B-Instruct with PULSE loss on UltraFeedback and HH-RLHF dataset. In the experiment, we used both original queries and responses of UltraFeedback and HH-RLHF as preference data.

Table 6 shows that PULSE yields performance gains across various benchmark datasets. The results highlight the effectiveness of our peer-review data construction framework: revision-based self-correction through textual feedback significantly enhances reasoning performance compared to prior methods.

Next, we compare PULSE trained on UltraFeedback with the original DPO, as shown in Table 7. PULSE consistently outperforms DPO, achieving gains of +15.6% on Alpaca LC, +12.9% on Alpaca 2.0, and +3.94% on MT-Bench. These results demonstrate the robustness and effectiveness of our proposed loss function.

Table 7: **Experimental results of Mistral-7B trained by the original Ultrafeedback dataset.**

| Model | Alpaca LC. | Alpaca 2.0 | MT-Bench |
|---|---|---|---|
| DPO | 18.59 | 17.42 | 7.10 |
| PULSE | **21.49** | **19.67** | **7.38** |

The robustness of the PULSE for preference learning can be explained in further detail as follows. The key strengths of PULSE in robustness lie in its two properties: boundedness of loss and its gradient and $\beta$-scheduling. Both properties contribute to stable and robust training dynamics. Our empirical observations show that PULSE maintains a more stable gradient norm compared to DPO, thereby reducing the risk of overly aggressive policy updates (see Appendix A.5 for detailed analysis).

Moreover, the primary goal of alignment is not necessarily to discover the global optimum in the optimization landscape as pursued during pretraining, but rather to identify a better local optimum that aligns with human preferences. From this perspective, PULSE provides a practical and effective alternative to DPO for preference-based LLM training.

**Weaker or Stronger reviewers.** We evaluate PULSE using critiques from both weaker and stronger LLMs. As shown in Table 8, PULSE-Gemma2-9B achieves a 16.5% gain on Alpaca LC and 2.8% on MT-Bench when guided by weaker models, supporting the notion of weak-to-strong generalization Burns et al. (2024). Despite their limited capabilities, weaker models provide useful feedback for effective self-correction. When using stronger critics, performance improves further, by 21.3% on Alpaca LC and 3.6% on MT-Bench.

Table 8: **Performance with weaker or stronger critic LLMs for Gemma2-9B.** Weaker critics are Qwen2.5-1.5B, LLaMA3.1-1B, and Gemma2-2B. Stronger critics are Qwen2.5-32B, Mistral-24B, and Gemma2-27B.

| Critic LLMs | Alpaca LC. | MT Bench |
|---|---|---|
| Gemma2-9B | 45.08 | 8.52 |
| w/ Weaker | 52.53 (+7.45) | 8.76 (+0.24) |
| w/ Stronger | 54.66 (+9.58) | 8.83 (+0.31) |

These findings confirm that PULSE supports collaborative training through LLM feedback, regardless of the critic's strength, and offers a versatile path toward scalable self-improving AI systems.

**Additional experiments.** Qualitative results are provided in Appendix A.16. We provide additional experiments on iterative learning (A.3), multi-round revision (A.4), stability of training (A.5), and inference cost analysis (A.10).

## 4 RELATED WORK

**LLM Self-Improving Methods.** Self-training approaches have emerged as an effective strategy to improve LLM performance by leveraging synthetic data generation and quality control mechanisms. Iterative DPO Dong et al. (2024); Xu et al. (2023) introduced a preference-based learning framework in which multiple responses are generated for a given prompt, and comparative judgments are constructed using either an LLM acting as a judge Bai et al. (2022), self-reward model Yuan et al. (2024); Ko et al. (2025), and a pre-trained reward model Jiang et al. (2023b); Liu et al. (2025). Alongside these developments, self-correction has been explored as an alternative approach for model refinement. STaR Zelikman et al. (2022) proposed a self-correction strategy for fine-tuning, encouraging LLMs to update their reasoning when provided with the correct answer. SCoRe Kumar et al. (2025) extended this concept and used reinforcement learning for improving upon their initial responses in subsequent attempts. However, despite their efficacy, the prior methods typically depend on access to ground-truth answers or reward models.

**LLM-based Multi-Agent Systems.** Recently, multi-agent collaboration systems based on LLMs have gained research interest, exploring the synergistic potential of cooperative language model agents Gao et al. (2023a); Xu et al. (2024); Zhou et al. (2024). The authors in Li et al. (2023a) investigated the mechanisms by which multi-agent LLM systems manage complex collaborative tasks, demonstrating emergent cooperative behaviors and theory-of-mind reasoning analogous to those observed in humans. In parallel, a collaborative evaluation Chan et al. (2024), and a reflective leveraging multi-LLM agents Bo et al. (2024) yield results that better align with human preferences. Wang et al. (2025) proposed the Mixture-of-Agents framework, which selects relevant agents from the agent pool to collaborate for the reasoning of the responses. These methods represent a pivotal direction for advancing the development of AI capable of operating without human supervision.

## 5 CONCLUSION

We propose PULSE, a novel learning paradigm based on peer review for collaborative LLM ecosystems. In PULSE, the actor LLM generates an initial response, which is revised based on feedback from peer critic LLMs. The initial and revised responses are evaluated by critics to construct preference datasets for the actor LLM to self-improve in an autonomous manner. PULSE presents a theoretically and empirically grounded noise-robust preference learning to address intrinsic challenges of learning from self-correction. PULSE achieved outstanding performance on various alignment and reasoning datasets, showing that it is a promising approach to LLM collaboration for self-improvement.

## 6 REPRODUCIBILITY STATEMENT

**Review process Implementation.** We experiment with various LLMs for the peer review process using vLLM. Those models can be accessed by Huggingface. We generate LLMs' responses using our prompt templates in Appendix A.11.

**Training Implementation.** We implement our training framework based on TRL. This framework provides trainers with various loss functions, but we develop proposed loss functions.

**Evaluation Datasets.** We evaluate Alpaca Eval and MT Bench, and other reasoning tasks using lm-evaluation-harness which are open-access repositories.

**Peer Review Dataset.** We attached the subset of preference datasets constructed by peer review in the supplementary material.

**Source Code.** We attached the source code in the supplementary material.

**Computational Resources.** All our experiments, we used one-eight A100 GPUs with 80GB VRAM.

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

# A APPENDIX

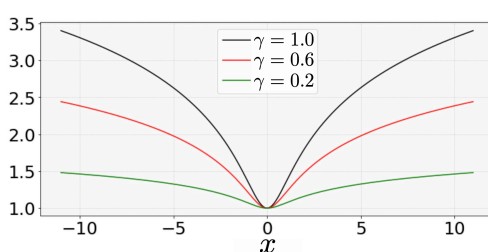
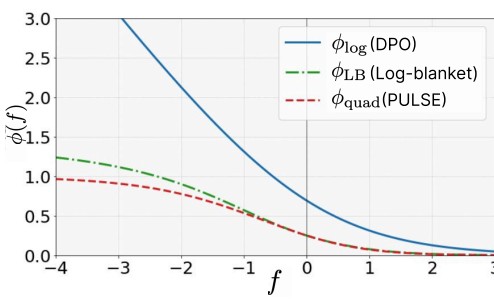

(a) Log-blanket function $1 + \frac{\gamma}{2}\log(1 + x^2)$ with different $\gamma$.

(b) Plots of $\phi_{\text{LB}}$ (Log-blanket), $\phi_{\text{quad}}$ (PULSE), and $\phi_{\text{log}}$ (DPO).

Figure 3: Log-blanket function and $\phi_{\text{LB}}$.

## A.1 DERIVATION OF (21).

Let us rewrite $L_{\text{PULSE}}(x, y; \theta)$:

$$\mathcal{L}_{\text{PULSE}}(x, y; \theta) = \frac{1}{\left[1 + \exp\left(\beta \log \frac{\pi_\theta(a_w|q)}{\pi_{\text{ref}}(a_w|q)} - \beta \log \frac{\pi_\theta(a_l|q)}{\pi_{\text{ref}}(a_l|q)}\right)\right]^2} \tag{22}$$

Let

$$c := \log \frac{\pi_\theta(a_w \mid q)}{\pi_{\text{ref}}(a_w \mid q)} - \log \frac{\pi_\theta(a_l \mid q)}{\pi_{\text{ref}}(a_l \mid q)}$$

which is constant for given $x, y, \theta$. Thus we have that, for $0 < \beta \ll |c|^{-1}$,

$$L_{\text{PULSE}}(x, y; \theta) := (1 + \exp(c\beta))^{-2} \tag{23}$$

$$= (1 + 1 + c\beta + o(\beta))^{-2} \tag{24}$$

$$= 2^{-2}\left(1 + \frac{c}{2}\beta + o(\beta)\right)^{-2} \tag{25}$$

$$= \frac{1}{4}\left(1 - c\beta + o(\beta)\right) \tag{26}$$

using the Taylor expansion. Similarly,

$$L_{\text{PULSE}}(x, -y; \theta) := (1 + \exp(-c\beta))^{-2} \tag{27}$$

$$= \frac{1}{4}\left(1 + c\beta + o(\beta)\right) \tag{28}$$

By combining (26) and (28), we have (21).

## A.2 LOSS WITH LOGARITHMIC RATE OF INCREASE.

We consider a loss function $\phi(f)$ which has logarithmic rate of increase when $f \to -\infty$. We propose a loss with *log-blanket* function as follows:

$$\phi_{\text{LB}}(f) = \underbrace{\left(1 + \frac{\gamma}{2}\log(1 + f^2)\right)}_{\text{log-blanket function}} \cdot \phi_{\text{quad}}(f)$$

The role of log-blanket is to cover (modulate) the envelope of the original loss $\phi_{\text{quad}}$ with a slowly increasing logarithm. Since $\phi_{\text{quad}} \to 1$ as $f \to -\infty$, $\phi_{\text{LB}}$ scales as $\sim \gamma \log|f|$ where $\gamma > 0$ is a hyperparameter. Thus, $\phi_{\text{LB}}$ is asymptotically concave, and its sensitivity to errors is between $\phi_{\text{log}}$ (linear) and $\phi_{\text{quad}}$ (constant). Examples of log-blanket functions and $\phi_{\text{LB}}$ are provided in Fig. 3.

Next, we examine the log-blanket loss from the perspective of risk minimization. It is difficult to find the minimum risk $C$ from $\phi_{LB}(f)$ using (9). However, $\phi_{LB}(f)$ has a desirable property as a loss for classification tasks as follows. Given some loss function $\phi$, suppose there exists $f_\phi^*$ that minimizes the conditional risk (8):

$$f_\phi^* = \arg\min_f \left[\mathbb{E}_{Y|x}[\phi(Yf(x))]\right] = \arg\min_f \left[\eta \cdot \phi(f) + (1-\eta) \cdot \phi(-f)\right]$$

We say $\phi$ is *Fisher consistent* if $f_\phi^*(x)$ has the same sign as $2\eta(x) - 1$, i.e., discriminant $f_\phi^*$ leads to a Bayes-optimal decision rule. The Fisher consistency of $\phi_{quad}$ for PULSE is implied because it is obtained from the risk minimization in (10) Zhang (2004). We show that the consistency holds for $\phi_{LB}$ as well.

**Proposition 1.** *Suppose $\gamma \leq \frac{\sqrt{10}-1}{9} \approx 0.241$. Then the loss $\phi_{LB}$ is Fisher consistent.*

*Proof.* (Lin, 2004, Theorem 3.1) provided a sufficient condition on loss $\phi$ for conditional risk $\mathbb{E}_{Y|x}[\phi(Yf(x))]$ to be Fisher consistent:

1. $\phi(z) < \phi(-z)$ for all $z > 0$

2. $\phi'(0) \neq 0$

3. $\mathbb{E}_{Y|x}[\phi(Yf(x))]$ attains a global minimum with minimizer $f^* \in \mathbb{R}$.

Then the sign of $f^*(x)$ is equal to $\eta(x) - \frac{1}{2}$, implying the Fisher consistency. We check the above three conditions. Let

$$g(z) = 1 + \frac{\gamma}{2}\log(1 + z^2)$$

denote the log-blanket function. We have that $\phi_{LB}(z) = g(z)\phi_{quad}(z)$. Condition 1 holds: for $z > 0$,

$$g(z)\phi_{quad}(z) < g(-z)\phi_{quad}(-z)$$

because $g(z) = g(-z)$ and $\phi_{quad}(z) < \phi_{quad}(-z)$.

For Condition 2, we have

$$\phi_{LB}'(z) = g'(z)\phi_{quad}(z) + g(z)\phi_{quad}'(z)$$

We have $g'(0) = 0$ and $g(0) = 1$, and thus $\phi_{LB}'(0) = \phi_{quad}'(0) \neq 0$

Next, we show Condition 3. We make the substitution

$$z(p) = \log\frac{p}{1-p}$$

for $p \in [0, 1]$. Note that $z(p)$ is the logit function whose range is $\mathbb{R}$. $z(p)$ is strictly increasing in $p$. Thus, finding the global minimizer for $\phi(z)$ over $z \in \mathbb{R}$ is equivalent to minimizing $\phi(z(p))$ over $p \in [0, 1]$. The log-blanket function $g$ is now given by the function of $p$:

$$g(p) := 1 + \frac{\gamma}{2}\log\left[1 + \left(\log\frac{p}{1-p}\right)^2\right] \tag{29}$$

We define the loss $\Phi(p) := \phi_{LB}(z(p))$ as

$$\Phi(p) = \underbrace{\left(1 + \frac{\gamma}{2}\log(1 + \left[\log\frac{p}{1-p}\right]^2)\right)}_{=g(p)} \cdot \underbrace{(\eta(1-p)^2 + (1-\eta)p^2)}_{:=h(p)} \tag{30}$$

We will show that $\Phi(p)$ is a convex function of $p$, which implies that $\Phi(p)$ attains a global minimum for $p \in [0, 1]$, and so does $\phi_{LB}(z)$ for $z \in \mathbb{R}$. With some algebra, one can show that log-blanket $g(p)$ is strictly convex in $p$, achieving the minimum value of 1 at $p = 0.5$. We consider the second derivative of $\Phi(p)$ with respect to $p$:

$$\Phi''(p) = \underbrace{g''(p)h(p)}_{(A)} + \underbrace{2g'(p)h'(p)}_{(B)} + \underbrace{g(p)h''(p)}_{(C)} \tag{31}$$

We will show that $\Phi''(p) \geq 0$. Firstly, (A) and (C) are nonnegative due to the convexity and nonnegativity of $g(p)$ and $h(p)$. Next, we examine (B) given by

$$\frac{\gamma z(p)}{1 + z^2(p)} \cdot \frac{4(p - \eta)}{p(1 - p)} \tag{32}$$

Assume $\eta \geq 0.5$ without loss of generality[2]. It is easy to check that $g'(p)$ and $h'(p)$ have the same sign if $p < 0.5$ or $p > \eta$, which implies (32) is nonnegative, or $\Phi''(p) > 0$ for $p < 0.5$ or $p > \eta$. Thus, we consider the interval $0.5 \leq p \leq \eta$ for which (32) is negative. We can bound (32) from below as

$$\frac{\gamma z(p)}{1 + z^2(p)} \cdot \frac{4(p - \eta)}{p(1 - p)} \geq \frac{\gamma z(p)}{1 + z^2(p)} \cdot \frac{4(p - 1)}{p(1 - p)} = -\frac{z(p)}{1 + z^2(p)} \cdot \frac{4\gamma}{p}$$

The first inequality is because, if we view the leftmost term as a function of $\eta$, it is decreasing in $\eta$ because it is multiplied by a negative number ($z(p) \geq 0$ for $p \in [0.5, \eta]$). Thus, we replace $\eta$ by 1 in the leftmost term to obtain the inequality.

Next, consider (A) in (31):

$$\underbrace{\frac{\gamma \left[(2p - 1)(1 + z^2(p))z(p) - z^2(p) + 1\right]}{(1 - p)^2 p^2 (1 + z^2(p))^2}}_{=g''(p)} \cdot \underbrace{(p^2 + \eta(1 - 2p))}_{h(p)} \tag{33}$$

Similarly, if we view (33) as a function of $\eta$, it is minimized at $\eta = 1$ considering $p \in [0.5, \eta]$ and the convexity of $g(p)$ or $g''(p) \geq 0$. Thus, (33) is bounded below by

$$\frac{\gamma \left[(2p - 1)(1 + z^2(p))z(p) - z^2(p) + 1\right]}{(1 - p)^2 p^2 (1 + z^2(p))^2} \cdot (p^2 - 2p + 1) = \frac{\gamma \left[(2p - 1)(1 + z^2(p))z(p) - z^2(p) + 1\right]}{p^2 (1 + z^2(p))^2}$$

Finally, (C) in (31) is $2g(p)$ which is bounded below by 2.

By combining the lower bounds on (A), (B) and (C), we obtain the following lower bound on $\Phi(p)$:

$$\Phi''(p) \geq \underbrace{\frac{\gamma \left[(2p - 1)(1 + z^2(p))z(p) - z^2(p) + 1\right]}{p^2 (1 + z^2(p))^2} - \frac{z(p)}{1 + z^2(p)} \cdot \frac{4\gamma}{p} + 2}_{:=G(p)}$$

$$\geq \frac{(1 + z^2(p)) \overbrace{\left[2p^2(1 + z^2(p)) - \gamma(2p + 1)z(p) - \gamma\right]}^{} + 2\gamma}{p^2(1 + z^2(p))^2}$$

We will show that $G(p) \geq 0$ for $p \in [0.5, \eta]$. The following holds for $p \in [0.5, \eta]$:

$$p^2 \geq \frac{1}{4}, \quad -(2p + 1) \geq -3$$

Applying these inequalities to $G(p)$, we have that

$$G(p) = 2 \underbrace{p^2}_{\geq \frac{1}{4}} (1 + z^2(p)) \underbrace{-\gamma(2p + 1)}_{\geq -3\gamma} z(p) - \gamma \geq \underbrace{\frac{z^2(p)}{2} - 3\gamma z(p) + \frac{1}{2} - \gamma}_{:=h(z)}$$

A sufficient condition on $G(p) \geq 0$ is $h(z) \geq 0$ where we view $h(z)$ as a quadratic function in $z$. By setting $h(z) = 0$ as a quadratic equation in $z$, we consider the condition that the discriminant of the quadratic equation is negative, in which case $h(z) \geq 0$ holds for any real $z$. The condition of the discriminant of $h(z) = 0$ being negative is given by

$$9\gamma^2 + 2\gamma - 1 \leq 0$$

This condition is met if

$$\gamma \leq \frac{\sqrt{10} - 1}{9} \approx 0.241$$

in which case we have $G(p) \geq 0$. Thus, $\Phi''(p) > 0$ for all $p \in [0, 1]$, and thus $\Phi(p)$ is strictly convex and attains a unique global minimum for $p \in [0, 1]$. This implies that $\phi_{\mathrm{LB}}(z)$ attains the global minimum for $z \in \mathbb{R}$, or Condition 3 is satisfied. $\qquad\square$

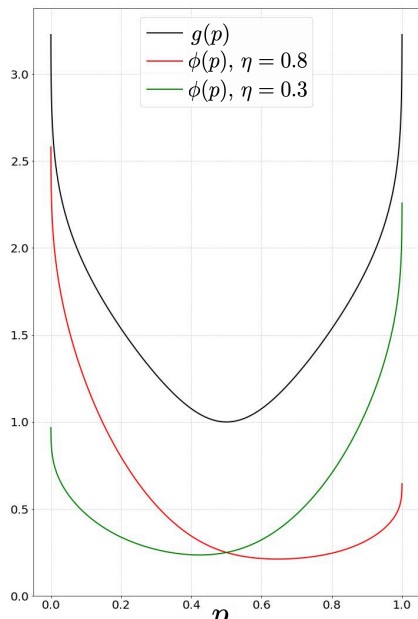

Figure 4: Visualization of $g(p)$ and $\Phi(p)$ functions with different $\eta$.

Table 9: Performances of Mistral-7B and LLaMA3-8B Alpaca LC. using the log-blanket loss with $\gamma = 0.2$.

| Loss $\mathcal{L}$ | Mistral-7B | LLaMA3-8B | Qwen2.5-7B | Gemma2-9B |
|---|---|---|---|---|
| Log-Blanket-PULSE | **29.86** | **27.54** | **34.53** | **54.46** |
| PULSE | 29.54 | 27.39 | 33.84 | 53.85 |

The upper bound 0.241 on $\gamma$ turns out to be loose, and our numerical study shows that $\phi_{LB}(p)$ remains convex for $\gamma$ over 2. The plots of log-blanket function $g(p)$ in (29) and log-blanket loss $\Phi(p)$ in (30) as a function of $p \in [0, 1]$ for $\gamma = 1$ are shown in Fig. 4. We observe that $g(p)$ and $\Phi(p)$ are strictly convex functions with unique global minima.

Overall, the log-blanket loss $\phi_{LB}$ is a simple modification of the original loss $\phi_{quad}$ with higher responsiveness to training errors and has a desirable property for classification.

Table 9 shows the experimental results comparing the original PULSE and the log-blanket PULSE using $\phi_{LB}$. We observe that the performance increases with the log-blanket loss, which shows that there is room for improvement by exploring the loss functions with various rates of change.

### A.3 ITERATIVE LEARNING.

We assess PULSE in an iterative learning setup using three rounds of UltraFeedback, following the Iterative DPO/IPO setting. As shown in Fig. 6, PULSE yields substantial improvements on Alpaca LC and AlpacaEval 2.0 in the first round, with continued gains in subsequent rounds, albeit smaller. In the final round, PULSE slightly surpasses the baseline trained on all prompts in a single pass, indicating its ability to support continual self-improvement and collaborative learning for LLMs.

### A.4 EXPERIMENTS ON MULTI-ROUND REVISIONS.

We also experimented with a multi-round revision process for peer review. As shown in Fig. 5, conducting more than one revision round did not yield significant improvements or show slight

---

[2]This assumption suffices for the proof due to the symmetry in $\Phi$: if we view $\Phi(p)$ as a function of both $p$ and $\eta$, or $\Phi(p, \eta)$, then $\Phi(p, \eta) = \Phi(1 - p, 1 - \eta)$.

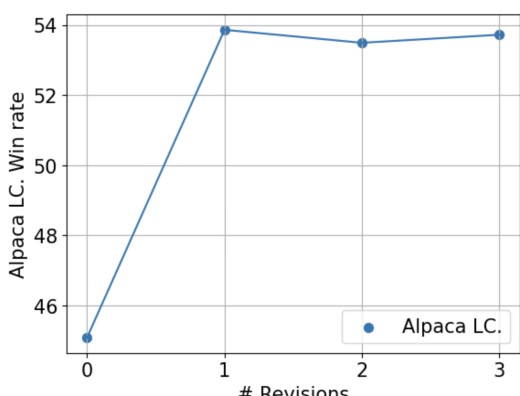

Figure 5: Performance improvement of Gemma2-9B per the number of revisions.

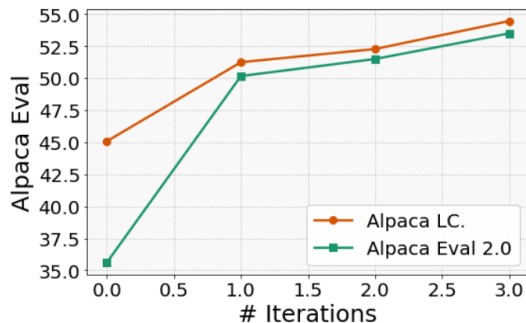

Figure 6: **Performance of Alpaca LC and MT-Bench for Gemma2-9B across iterative learning.** We use the same critic LLMs as in our main experiments, and employ a three-round curated Ultra-Feedback dataset.

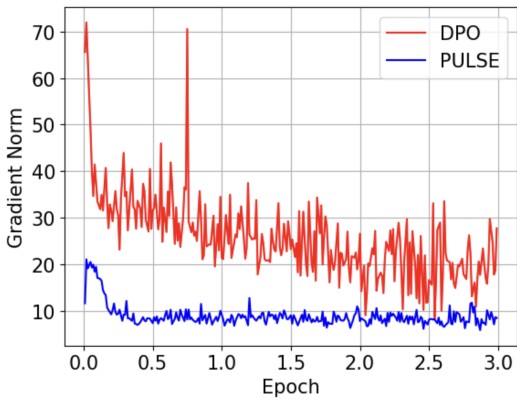

Figure 7: Gradient norm of Qwen2.5-7B during training from DPO vs. PULSE.

performance degradation. This is likely because LLMs tend to address most of their weaknesses during the initial revision, while subsequent rounds primarily introduce noise, as current small LLMs are not yet sufficiently robust to handle iterative peer review processes effectively. Nevertheless, as AI models continue to evolve, multi-round revisions may become more effective and even necessary for tackling more complex tasks such as proving mathematical theorems or solving harder reasoning problems in the future.

| Method | Alpaca LC. | GSM8K | MATH | HumanEval | MBPP | MMLU |
|---|---|---|---|---|---|---|
| PULSE w/o text feedback | 25.34 | 45.92 | 9.22 | 9.76 | 31.40 | 55.78 |
| PULSE w/ text feedback | **29.54** | **47.61** | **12.70** | **17.68** | **37.80** | **57.53** |

Table 10: Ablation study showing the contribution of PeerReview data and $\mathcal{L}_{PULSE}$.

| Method | Alpaca LC. | GSM8K | MATH | HumanEval | MBPP | MMLU |
|---|---|---|---|---|---|---|
| DPO w/ Skywork RM | 27.45 | 45.13 | 10.25 | 8.72 | 19.24 | 55.62 |
| PULSE w/ Skywork RM | **28.59** | **46.15** | **10.68** | **10.56** | **21.78** | **56.56** |

Table 11: Ablation study comparing DPO and $\mathcal{L}_{\text{PULSE}}$ trained on Skywork RM data.

## A.5    EFFECT OF LOSS FUNCTION ON GRADIENT UPDATES.

We compare the stability of training DPO and PULSE by analyzing the change in gradient norms over the training steps. As shown in Fig. 7, DPO exhibits large and highly fluctuating gradient norms, with peaks reaching over 70 in the early training phase. This indicates unstable parameter updates, which can lead to inefficient or even divergent training dynamics. In contrast, our proposed method gradually increases the gradient norm up to 20 and consistently has low gradient norms, stabilized around 10 throughout the entire training process. This suggests that the model performs more refined and controlled updates, likely contributing to a smoother optimization trajectory.

The gradual increase of the gradient, similar to curriculum learning, mitigates the overfitting of undesirable local minima in early training. In addition, the significantly reduced gradient magnitude and variance further indicate that our method effectively mitigates the risk of gradient explosion and leads to steady convergence. This stability is often correlated with better generalization, suggesting that our approach yields more reliable and robust learning behavior.

## A.6    ABLATION STUDY OF TEXTUAL FEEDBACK

Table 10 shows the performance without the structured text feedback. For the ablation of text feedback, we removed the textual guidance and prompt the actor LLM only to "improve your previous answer". We observe that the performance drops substantially across all benchmarks (e.g., GSM8K 47.61 → 10.92, HumanEval 17.68 → 9.76, Alpaca LC 29.54 → 25.34). This shows that text feedback is crucial. The actor does not benefit merely from generic self-correction instructions. Instead, **structured feedback from peers provides explicit and high-quality guidance** on what was wrong and how to improve. This aspect is essential to the performance gain of PULSE.

## A.7    ADDITIONAL ABLATION: COMPARISON WITH DPO ON SKYWORK RM DATA

This ablation examines whether the proposed $\mathcal{L}_{\text{PULSE}}$ offers advantages over DPO when both are trained on the same Skywork Reward Model (RM). Using identical student models, prompts, and RM annotations, we isolate the effect of each objective. As shown in Table 11, $\mathcal{L}_{\text{PULSE}}$ consistently surpasses DPO across all benchmarks, with especially notable gains in code generation (HumanEval, MBPP) and instruction following (Alpaca LC.). These results demonstrate that even under identical preference supervision, the optimization mechanism significantly impacts final performance, and PULSE leverages preference data more effectively than DPO, yielding stronger and more reliable alignment.

## A.8    DISCUSSION ON PRIOR NOISE-ROBUST METHODS.

Recently, some noise-robust variants of DPO, e.g., rDPO Chowdhury et al. (2024), cDPO Mitchell (2023), have been proposed. The variants rely on the following key assumptions:

- Each sample is mislabeled at random with probability $\epsilon$.
- Probability $\epsilon$ is either known in advance or can be estimated reasonably accurately.

These requirements are explicitly stated in rDPO paper Chowdhury et al. (2024):

*We particularly focus on the DPO algorithm in the presence of random preference noise, where preferences are flipped with some (known) rate.*

These variants explicitly use the error rate $\epsilon$ in their algorithms, while our method does not need $\epsilon$. Specifically, consider the rDPO loss Chowdhury et al. (2024) and the cDPO loss Mitchell (2023):

$$\textbf{(rDPO loss)} = \frac{\epsilon \mathcal{L}_{\text{DPO}}(q, a_w, a_l) - (1 - \epsilon)\mathcal{L}_{\text{DPO}}(q, a_l, a_w)}{1 - 2\epsilon}$$

$$\textbf{(cDPO loss)} = \epsilon \mathcal{L}_{\text{DPO}}(q, a_w, a_l) + (1 - \epsilon)\mathcal{L}_{\text{DPO}}(q, a_l, a_w)$$

where $\mathcal{L}_{\text{DPO}}(q, a_w, a_l)$ is the standard DPO loss for query $q$ and preferred response $a_w$ and dispreferred response $a_l$. The key assumption for these losses is that the probability of label error $\epsilon$ is independent of query $q$ or responses $a_w, a_l$. The assumption may not hold in practice, i.e., label errors are unlikely to occur at random. Instead, the error likely depends on the context of the samples, such as the ambiguity of the contexts or the subjectivity of annotators. Moreover, $\epsilon$ is difficult to know or accurately estimate in practice. Human annotators or strong LLMs are needed to examine the samples, which is difficult to do at scale.

In conclusion, we believe that rDPO and its variants are valuable studies on robustness under idealized assumptions on error distribution. However, their performance may not be guaranteed for practical preference datasets, such as our dataset.

## A.9   OTHER PREFERENCE LOSSES AND RISK MINIMIZATION.

We searched for other preference losses to which we can apply risk minimization. From recent works, we found the losses of IPO Azar et al. (2024)

$$\mathcal{L}_{\text{IPO}} = (\delta - (r_w - r_l))^2$$

and SLiC Zhao et al. (2023)

$$\mathcal{L}_{\text{SLiC}} = \max(0, \delta - (r_w - r_l))$$

These losses have the following in common: $r_w$ denotes the reward for the winning response and $r_l$ denotes the reward for the losing response. Now we can apply the risk minimization framework from Zhang (2004) to find the corresponding losses and minimum conditional risks. We provide a summary of the comparison of these losses in Table A.9.

| Loss | $C(p)$ | $\phi(f)$ | Key Property ($f \to -\infty$) |
|---|---|---|---|
| DPO Rafailov et al. (2023) | $-p \log p - (1 - p) \log(1 - p)$ | $-\log \sigma(f)$ | Unbounded, nondecreasing gradient, sensitive to noise |
| IPO Azar et al. (2024) | $p(1 - p)$ | $(1 - f)^2$ | Unbounded, nondecreasing gradient, sensitive to noise |
| SLiC Zhao et al. (2023) | $1 - |2p - 1|$ | $\max(1 - f, 0)$ | Unbounded, nondecreasing gradient, sensitive to noise |
| PULSE (Ours) | $p(1 - p)$ | $(1 - \sigma(f))^2$ | Bounded, decreasing gradient, robust to noise |

Table 12: Comparison of preference losses via their minimum conditional risk $C(p)$.

## A.10   INFERENCE COST ANALYSIS

PULSE requires inferences from the author and peer LLMs to generate, review and revise the responses. However, PULSE is not computationally expensive, as it has only two rounds of the generation process. In each round, a model generates a response, followed by 3 independent reviewers producing critiques, resulting in 4 LLM generations per round, or 8 in total per example. Considering that recent costs of LLM inference has dropped significantly, PULSE offers a scalable approach to constructing high-quality preference data.

Next, we compare the computation cost of PULSE versus the human annotation fee. Our measurement shows that PULSE requires 115.5M tokens to construct preference datasets for Mistral-7B. To simplify cost estimation, we used batch API pricing of GPT-4.1-nano ($0.2 per 1M tokens), because 7B-scale models are comparable to GPT-4.1-nano in capability. In this case, the estimated cost of PULSE is only $23.1.

Next, we estimate the cost of human annotation. A simple annotation of text data can range between $0.1-$0.5 per sample. If we set the average cost per sample as $0.25, the human cost is estimated at $14,032.8. In conclusion, the inference cost of PULSE is negligible at 0.16% of the human annotation cost.

### A.11 PROMPT TEMPLATES

---

**Listing 1** Prompt used for initial review.

```
f'''
Your task is to review the peer's answer to the question below. After
↪   reading and thinking deeply about the answer, you can rate the grade
↪   between 1 and 5, whether the current answer is acceptable.
For example, 1: reject, 2: weak reject, 3: borderline, 4: weak accept,
↪   5: accept.
You should write both the summary of strengths and weaknesses about the
↪   answer, including what needs to be revised. When you are ready to
↪   write, conclude using the format Score: "..."\nWeaknesses:
↪   "..."\nStrengths: "..."\nRecommended Suggestions: "...".
Question: {}
Answer: {}
'''
```

---

**Listing 2** Prompt used for final review.

```
f'''
Your task is to review the peer's answer to the question below. After
↪   reading and thinking deeply about the answer, you can rate the grade
↪   between 1 and 5 on whether the current answer is acceptable.
For example, 1: reject, 2: weak reject, 3: borderline, 4: weak accept,
↪   5: accept.
You should write the reason why you rated your score. When you are ready
↪   to write, conclude using the format Score: "..."\nReason: "...".
Question: {}
Answer: {}
'''
```

---

**Listing 3** Prompt used for revision.

```
f'''
Your task is to revise your previous answer based on the peer's feedback.
↪   After reading and thinking deeply about the review, you can correct
↪   your previous answer by referring to the weaknesses and suggestions
↪   while retaining the strengths.
Question: {}
Previous Your Answer: {}
Feedback: {}
'''
```

---

## A.12 HYPERPARAMETERS

Table 13: Detailed hyperparameters.

| Hyperparameter | Value |
|---|---|
| Batch Size | 64 |
| Learning rate | 1e-7 |
| Learning rate scheduler | Linear |
| Max Prompt Length | 512 |
| Max Sequence Length | 512 |
| Epochs | 3 |
| Optimizer | RMSProp |
| Warmup ratio | 0.1 |
| $\beta$ | 0.05, 0.1 |
| $\lambda$ | 0.1 |
| $\delta$ | $\dfrac{1}{0.03 \times (\text{total training steps})}, \dfrac{1}{0.05 \times (\text{total training steps})}$ |

## A.13 K-SHOTS IN REASONING TASKS.

For evaluating reasoning tasks, we use the default few-shot settings for each dataset as implemented in lm-evaluation-harness.

Table 14: The number of shots for evaluating each reasoning task.

| | |
|---|---|
| GSM8K | 5-shot |
| MATH | 4-shot |
| HumanEval | 0-shot |
| MBPP | 0-shot |
| GPQA-Diamond | 5-shot |
| MMLU | 5-shot |

## A.14 EVALUATION SETTINGS OF ALPACA EVAL AND MT-BENCH

**LLM-Judge:** We used GPT-4 as the automated judge. Specifically, we employed the `gpt-4-1106` model, which is the recommended and widely used model in the LLM-as-a-judge framework for stable and reproducible scoring.

**Decoding Parameters:** For generating model responses submitted to the judge, we used default evaluation parameters balancing quality and diversity:

- Temperature: 0.7
- Top-p: 1.0
- Max New Tokens: 2048

A.15 EXAMPLES OF PEER REVIEW PROCESS

Figure 8: **Example of review process of LLaMA-8B.**

**Prompt:** Q: Given a scientific question, generate a correct answer to it. When water particles become hot enough to overcome the force of attraction between them, what happens to the water?
A:

**Initial response of LLaMA-8B:**
When water particles become hot enough to overcome the force of attraction between them, they change from a liquid to a gas, a process known as vaporization. This occurs when the kinetic energy of the water molecules increases and they begin to move faster and faster, eventually breaking free from the intermolecular forces that hold them together in a liquid state. As a result, the water vaporizes and turns into steam.

**Review 1:**
Score: 4 (Weak Accept)
Weaknesses: The answer is generally strong but could be more precise and detailed. The explanation of the process is accurate, but it could benefit from a more explicit mention of the specific type of vaporization (boiling) and the conditions under which it occurs.
Strengths: The answer correctly identifies that the process is vaporization and provides a clear explanation of the molecular behavior leading to this change. It also mentions the increase in kinetic energy and the breaking of intermolecular forces, which are key concepts in understanding the transition from liquid to gas.
Sugesstions: 1) Specify that this process is specifically called "boiling" and occurs at the boiling point of water (100°C or 212°F at standard atmospheric pressure). 2) Clarify that the water changes from a liquid to a gas at the surface and throughout the liquid, not just at the surface. 3) Mention that the process can also occur at temperatures below the boiling point through a process called evaporation, which happens at the surface of the liquid.

**Review 2:**
Score: 5 (Accept)
Weaknesses: None
Strenghts: The answer is accurate, concise, and clearly explains the process of vaporization. It correctly identifies the key concepts involved, such as the force of attraction between water particles, kinetic energy, and intermolecular forces.
Suggestions: The answer is already well-written and does not require any corrections.

**Review 3:**
Score: 4 (Weak Accept)
Weakness: The answer is generally correct, but it could be more specific about the temperature at which water vaporizes. Also, the explanation could be clearer about what causes the kinetic energy of the water molecules to increase.
Strengths: The answer correctly identifies the process of vaporization and explains that it occurs when the intermolecular forces are overcome.
Suggestions: To improve the answer, it would be helpful to specify that water vaporizes at 100 degrees Celsius (212 degrees Fahrenheit) at standard atmospheric pressure. Additionally, the explanation could be revised to clarify that the increase in kinetic energy is due to the heat energy being absorbed by the water molecules.

**Revision:**
When water particles become hot enough to overcome the force of attraction between them, they undergo a process called boiling, which is a type of vaporization. This occurs when the kinetic energy of the water molecules increases due to the absorption of heat energy, causing them to move faster and faster and eventually break free from the intermolecular forces that hold them together in a liquid state. As a result, the water changes from a liquid to a gas, turning into steam. This process occurs at the boiling point of water, which is 100°C (212°F) at standard atmospheric pressure. Importantly, boiling does not only occur at the surface of the liquid but can also happen throughout the liquid. Additionally, water can also vaporize at temperatures below the boiling point through a process called evaporation, which occurs at the surface of the liquid.

I have made the following changes to address the weaknesses and recommended suggestions:
1. I specified that the process is called "boiling" and occurs at the boiling point of water (100°C or 212°F at standard atmospheric pressure).
2. I clarified that the water changes from a liquid to a gas throughout the liquid, not just at the surface.
3. I mentioned that the process can also occur at temperatures below the boiling point through evaporation.
4. I rephrased the explanation to clarify that the increase in kinetic energy is due to the absorption of heat energy.
# Self-Reflection

Figure 9: **Example of review process of Mistral-7B.**

**Prompt:** What is chain rules in calculus.

**Initial response of Mistral-7B:**
The Chain Rule in calculus is a fundamental rule for differentiating composite functions. It allows us to find the derivative of a composite function by breaking it down into simpler functions.
The Chain Rule states that if we have a composite function f(g(x)), where f is an outer function and g is an inner function, then the derivative of the composite function can be found by the following formula:
$(f(g(x)))' = f'(g(x)) * g'(x)$
In other words, to differentiate a composite function, take the derivative of the outer function, evaluate it at the inner function, and multiply it by the derivative of the inner function. This rule is incredibly useful for finding derivatives of complex functions that can be broken down into simpler ones.

**Review 1:** ...
Weaknesses: The answer is generally clear and concise, but it lacks specific examples to illustrate the application of the Chain Rule. Additionally, the explanation could be more detailed and rigorous, particularly in the definition of the Chain Rule.
Strengths: The answer provides a clear and concise definition of the Chain Rule, and it explains the formula in a straightforward manner. The language is easy to understand, and the answer is well-organized.
Suggestions: * Provide specific examples to illustrate the application of the Chain Rule, such as differentiating a function like $\sin(x^2)$ or $e^{(x^2)}$.
* Expand on the definition of the Chain Rule, providing more mathematical rigor and justification for the formula.
* Consider adding a visual representation, such as a diagram or graph, to help illustrate the concept of the Chain Rule.
**Review 2:** ...
Weaknesses: The answer is generally clear and well-structured, but it could benefit from more detailed explanation ...
**Review 3:** ...
**Suggestions:** * Include a brief example demonstrating how to use the Chain Rule to find the derivative of a composite function. For instance, find the derivative of $f(g(x)) = \sin(x^2)$ where $f(x) = \sin(x)$ and $g(x) = x^2$.

**Revision:** The Chain Rule in Calculus is a fundamental rule for differentiating composite functions. It allows us to find the derivative of a composite function by breaking it down into simpler functions.
The Chain Rule states that if we have a composite function f(g(x)), where f is an outer function and g is an inner function, then the derivative of the composite function can be found by the following formula:
(f(g(x)))' = f'(g(x)) * g'(x)
In other words, to differentiate a composite function, take the derivative of the outer function, evaluate it at the inner function, and multiply it by the derivative of the inner function. This rule is incredibly useful for finding derivatives of complex functions that can be broken down into simpler ones.
Let's take a look at some examples:
1. Consider h(x) = $(3x^2 + 2)^4$. Here, the outer function is $f(u) = u^4$ and the inner function is $g(x) = 3x^2 + 2$.
a. Differentiate the outer function f(u) with respect to u:

$$f'(u) = 4u^3$$

b. Evaluate f'(u) at the inner function g(x):

$$f'(g(x)) = 4(3x^2 + 2)^3$$

c. Differentiate the inner function g(x) with respect to x:

$$g'(x) = 6x$$

d. Multiply the results from steps b and c:

$$h'(x) = f'(g(x)) \cdot g'(x) = 4(3x^2 + 2)^3 \cdot 6x = 24x(3x^2 + 2)^3$$

2. Another example is finding the derivative of $f(g(x)) = \sin(x^2)$, where $f(x) = \sin(x)$ and $g(x) = x^2$.
a. Differentiate the outer function f(x) with respect to x:

$$f'(x) = cos(x)$$

b. Differentiate the inner function g(x) with respect to x:

$$g'(x) = 2x$$

c. Apply the Chain Rule:
$$f'(g(x)) \cdot g'(x) = cos(x^2) \cdot 2x$$

Figure 10: **Example of review process of Qwen2.5-7B.**

**Prompt:** In addition to computing the sum of all numbers that are divisible by 3 or 5, you are required to identify the index positions of those numbers within the original string. Please ensure that the index positions are also included in the final output.

**Initial response of Qwen2.5-7B:**
Sure, I can help you with that. Can you please provide me with more details on the original string and the range of numbers to consider? Also, do you want the index positions to be 0-based or 1-based?

**Review 1:**
Score: 2 (Weak Reject)
Weaknesses: The answer does not directly address the problem and instead asks for additional information, which is not relevant to the problem. The answer does not provide a clear solution to the problem and does not demonstrate an understanding of the requirements.
Strengths: The answer shows an attempt to engage with the problem by asking for more details, which is a good starting point. However, the answer does not provide a clear direction or solution.
...
**Review 2:**
Score: 2 (Weak Reject) ...
**Review 3:**
... Suggestions: To provide a complete solution, the answer should include a step-by-step approach to compute the sum of the numbers divisible by 3 or 5 and identify their index positions within the original string. The approach should be clear and concise, and should include any necessary assumptions about the original string and the range of numbers to consider. Additionally, the answer should specify whether the index positions are to be 0-based or 1-based.

**Revision:**
To solve the problem, we need to compute the sum of all numbers that are divisible by 3 or 5 within a given string and also identify the index positions of those numbers. Let's break down the solution step-by-step.
### Step-by-Step Solution
1. **Initialize Variables:**
- 'sum_divisible': To store the sum of numbers divisible by 3 or 5.
- 'index_positions': To store the index positions of the numbers divisible by 3 or 5.
2. **Iterate Through the String:**
- Convert each character in the string to an integer. - Check if the number is divisible by 3 or 5. - If it is, add the number to 'sum_divisible' and record its index position in 'index_positions'.
3. **Return the Result:**
- Return the sum and the list of index positions.
### Example
Let's consider the string ''123456'' as an example.
1. **Initialize:**
- 'sum_divisible = '
- 'index_positions = []'
2. **Iterate Through the String:**
- Convert each character to an integer and check divisibility:
- '1' (not divisible by 3 or 5)
...
3. **Final Output:**
- 'sum_divisible = 3 + 5 + 6 = 14'
- 'index_positions = [2, 4, 5]'

### Code Snippet
Here is a Python code snippet that implements the above logic:
```python
def sum_and_indices(s):
    sum_divisible = 0
    index_positions = []

    for i, char in enumerate(s):
        num = int(char)
        if num % 3 == 0 or num % 5 == 0:
            sum_divisible += num
            index_positions.append(i)

    return [sum_divisible, index_positions]
```

Figure 11: **Example of review process of Gemma2-9B.**

**Prompt:** aubrey got paid handsomely so he finished lee's paper for him. Given the context: What will Lee want to do next?
Answer: thank aubrey

Quinn recently made a friend and like to play soccer with them. Given the context: What will Quinn want to do next?
Answer: spend more time with their friend

Addison threw a punch that almost hit Remy, but they missed. Given the context: How would Remy feel as a result?
Answer: upset

Jesse was holding his test to turn in and dropped it into the crack between his desk. Given the context: How would you describe Jesse?
Answer:

**Initial response of Gemma2-9B:**
Answer: **disappointed or frustrated**
Let me know if you'd like to try another one!

**Review 1:**
Score: 3 (Borderline)
Weaknesses: The answer lacks specific details and context from the given situation. The answers seem to be very general and do not take into account the specific circumstances of the scenario.
Strengths: The answers are somewhat relevant to the given context, but could be improved with more specific details and analysis.
...
**Review 2:**
Score: 4 (Weak Accept)
Weaknesses: The answer provided is somewhat generic and does not fully address the context or the specific situation described. It lacks depth and specificity, which could lead to a more nuanced understanding of Jesse's emotional state.
Strengths: The answer is clear and directly addresses the question, providing a reasonable emotional response based on the context given.
Suggestions: To strengthen the answer, consider adding more context or specific details that reflect Jesse's situation. For example: "Jesse would likely feel disappointed and frustrated, as he has to retrieve his test and may be worried about the consequences of losing it.
**Review 3:**
Score: 3 (Borderline)
Weaknesses: The answer is not specific enough, as it does not directly address how Jesse might feel immediately after dropping the test.
Strengths: The answer correctly identifies a possible emotion that Jesse might feel in this situation.
Suggestions: To improve the answer, it would be beneficial to specify the immediate emotion Jesse might feel upon dropping the test, such as "frustrated" or "disheartened." A more detailed response could also include a description of Jesse's actions or thoughts following the drop, such as "Jesse might feel disappointed and frustrated, and he might start to search for the test or consider asking the teacher for an extension."

**Revision:** Jesse would likely feel **disheartened and frustrated** the moment he realizes his test has fallen into the crack. He might frantically try to retrieve it, worrying about the consequences of losing it and the potential impact on his grade.
Here's how I revised the answer based on the feedback:
* **Specificity:** Instead of just "disappointed or frustrated," I used more specific emotions like "disheartened" and "frustrated" to capture the immediate feeling upon dropping the test.

* **Action and Thought:** I added details about Jesse's possible actions ("frantically try to retrieve it") and thoughts ("worrying about the consequences") to show a more nuanced emotional response.

* **Contextualization:** I emphasized the potential impact of losing the test on Jesse's grade, grounding the emotions in the specific situation. # Self-Reflection

A.16   QUALITATIVE ANALYSIS

Figure 12: **Example of Mistral-7B responses in AlpacaEval 2.0.**

---

**Prompt:** Provide a formula for computing the nth term in the given sequence.
5, 14, 23, 32, 41, ...

---

**Mistral-7B base response:**
The given sequence appears to be an arithmetic sequence with a common difference of 9. To find the formula for the nth term, we can use the following formula:
an = a1 + (n-1)d
where:
an = nth term
a1 = first term
n = term number
d = common difference

Using the given sequence, we have:
a1 = 5
d = 9

Substituting these values into the formula, we get:
an = 5 + (n-1)9
an = 5 + 9(n-1)
an = 5 + 9n - 9
an = 9n + 4
So, the formula for the nth term of the given sequence is an = 9n + 4.

---

**DPO response:**
The given sequence appears to be an arithmetic sequence where the difference between each term is increasing by 9.
To find the common difference (d), we can subtract the first term from the second term:
d = 14 - 5 = 9
To find the formula for the nth term (An), we can use the formula for an arithmetic sequence:
An = a + (n - 1) * d
Where a is the first term, and d is the common difference. In this case, a = 5 and d = 9.
So, the formula for the nth term is:
An = 5 + (n - 1) * 9
To verify the formula, let's calculate the 6th term (A6) using the formula and compare it with the 6th term in the given sequence: # Self-Reflection
A6 = 5 + (6 - 1) * 9 = 5 + 5 * 9 = 49
The 6th term in the given sequence is 41 + 8 = 49, which matches our calculation.
Therefore, the formula for the nth term of the given sequence is An = 5 + (n - 1) * 9.

---

**PULSE response:**
The sequence you've provided appears to be an arithmetic sequence where each term increases by 9 (5 to 14, 14 to 23, etc.). To find the formula for the nth term, we can use the general formula for arithmetic sequences, a_n = a_1 + (n-1)d, where:
* a_n is the nth term of the sequence
* a_1 is the first term of the sequence (which is 5 in this case)
* n is the term number
* d is the common difference (which is 9 in this case)
So, the formula for the nth term of this sequence is:
a_n = 5 + (n-1) * 9
This formula will give you the nth term of the sequence. For example, a_5 = 5 + (5-1) * 9 = 41, which matches the given sequence.
You can verify that this formula works for any term in the sequence by substituting the term number into the formula. For example, a_1 = 5 (since the first term is 5), a_2 = 14, a_3 = 23, and so on.
You can also check the formula for larger values of n to ensure it works correctly. For example, a_10 = 5 + (10-1) * 9 = 59, which is the correct value for the 10th term in the sequence. # Self-Reflection
In general, this formula works for any arithmetic sequence where the difference between terms is constant. If the difference between terms varies, a different formula would be needed to find the nth term.

---

Figure 13: **Example of Mistral-7B responses in AlpacaEval 2.0.**

**Prompt:** Give some examples of what people usually say in the given social situation.
When someone arrives safely

**DPO response:**
1. "Welcome home! I'm so glad you made it back safely."
2. "I'm so relieved to see you safe and sound. It was a long journey."
3. "It's so good to see you back! I was worried about you. Safe travels!"
4. "Oh, thank goodness you're here. I was starting to worry. Are you okay?"
5. "I'm so happy to see you back in one piece. You must be exhausted. Take a rest."
6. "I can't believe you're back! I was so worried. Are you ready to share your adventure stories?"
7. "It's so good to see you back. I was starting to worry. I'm just so glad you're safe."
8. "Welcome back! I'm so relieved to see you. How was your trip?"
9. "I'm so glad you made it back safely. It must have been a tough journey."
10. "I'm so happy to see you back. I was starting to worry. Are you okay?"
11. "Welcome back! I'm so glad you're safe. I was starting to worry."
12. "I'm so happy to see you back. I was starting to worry. How was your trip?"
13. "Welcome back! I'm so glad you made it back safely. You must be exhausted. Take a rest."
14. "It's so good to see you back. I was starting to worry. Are you okay?"
15. "I'm so happy to see you back. I was starting to worry. How was your trip?"
16. "Welcome back! I'm so glad you made it back safely. It must have been a long journey."
17. "I'm so happy to see you back. I was starting to worry. Are you okay?"
18. "Welcome back! I'm so glad you made it back safely. How was your trip?"
19. "I'm so glad you made it back safely. It must have been a tough journey. Are you okay?"
20. "Welcome back! I'm so glad you made it back safely. It's good to have you back home."
21. "I'm so happy to see you back. I was starting to worry. How was your trip?"
22. "Welcome back! I'm so glad you made it back safely. You must be exhausted. Take a rest."
23. "I'm so glad you made it back safely. It must have been a long journey. Are you okay?"
24. "Welcome back! I'm so glad you made it back safely. How was your trip?"
25. "I'm so happy to see you back. I was starting to worry. Are you okay?"
26. "Welcome back! I'm so glad you made it back safely. It must have been a tough journey."
27. "I'm so glad you made it back safely. It must have been a long journey. Are you okay?"
28. "Welcome back! I'm so glad you made it back safely. How was your trip?"
29. "I'm so happy to see you back. I was starting to worry. Are you okay?"
30. "Welcome back! I'm so glad you made it back safely. It must have been a long journey."
31. "I'm so glad you made it back safely. It must have been a tough journey. Are you okay?"
32. "Welcome back! I'm so glad you made it back safely. How was your trip?"
33. "I'm so happy to see you back. I was starting to worry. Are you okay?"
34. "Welcome back! I'm so glad you made it back safely. It must have been a long journey."
35. "I'm so glad you made it back safely. It must have been a tough journey. Are you okay?"
36. "Welcome back! I'm so glad you made it back safely. How was your trip?"
37. "I'm so happy to see you back. I was starting to worry. Are you okay?"
38. "Welcome back! I'm so glad you made it back safely. It must have been a long journey."
39. "I'm so glad you made it back safely. It must have been a tough journey. Are you okay?"
40. "Welcome back

**PULSE response:**
1. "Welcome home! Safe journey."
2. "Thank goodness you're safe! I was worried sick."
3. "I'm so relieved to see you! It's great to have you back."
4. "I'm so glad you made it safely. It's good to have you back."
5. "It's a relief to see you're safe and sound."
6. "You're back! I'm so glad everything went well."
7. "I'm so happy you're here! It's a relief to know you're safe."
8. "Thank you for letting me know you arrived safely."
9. "I was on edge until I heard you were okay. It's good to see you."
10. "I'm so grateful you're safe. It's great to have you back with us."

Figure 14: **Example of LLaMA-8B responses in AlpacaEval 2.0.**

**Prompt:** Solve for x in the equation 3x + 10 = 5(x - 2).

**DPO response:**
To solve for x, I'll start by following the order of operations (PEMDAS) and distributing the 5 to the terms inside the parentheses:

3x + 10 = 5x - 10

Next, I'll add 10 to both sides to get all the constants on one side and the variables on the other:

3x = 5x - 20 + 10

This simplifies to:

3x = 5x - 10

Subtracting 5x from both sides gives me:

-2x = -10

Dividing both sides by -2 gives me:

x = 5

So, the value of x is 5!

**PULSE response:**
To solve for x, I'll follow the order of operations (PEMDAS) and rewrite the equation as:
3x + 10 = 5x - 10

Next, I'll add 10 to both sides of the equation to get:

3x = 5x - 20

Then, I'll subtract 5x from both sides to get:

-2x = -20

Finally, I'll divide both sides by -2 to solve for x:

x = 10

Therefore, the value of x is 10.

Figure 15: **Example of Qwen2.5-7B responses in AlpacaEval 2.0.**

**Prompt:** Change the response to have a more empathic tone in the chat.
+ How are you doing?
- Fine, I had a bunch of problems today.
+ What kind of problems?

**DPO response:**
+ How are you doing?
- I'm trying to manage, but I had a bunch of challenges today.
+ What kind of challenges are you dealing with? I'm here to listen if you want to talk about it.
-

**PULSE response:**

I'm really sorry to hear that you're having a tough day. What kind of problems are you dealing with? Sometimes just talking about them can help lighten the load a bit.

Figure 16: **Example of Gemma2-9B responses in AlpacaEval 2.0.**

**Prompt: Implement a queue data structure using two stacks in Python.**

**DPO response:**

```python
class Queue:
    def __init__(self):
        self.input_stack = []   # Used for enqueuing
        self.output_stack = []   # Used for dequeuing

    def enqueue(self, data):
        """Add an element to the rear of the queue."""
        self.input_stack.append(data)

    def dequeue(self):
        """Remove and return the element at the front of the queue."""
        if not self.output_stack:  # If output stack is empty
            while self.input_stack:  # Move all elements from input to output
                self.output_stack.append(self.input_stack.pop())
        if self.output_stack:
            return self.output_stack.pop()  # Dequeue from output stack
        else:
            return None   # Queue is empty

    def peek(self):
        """Return the element at the front of the queue without removing it."""
        if not self.output_stack:  # If output stack is empty, move elements from input
            while self.input_stack:
                self.output_stack.append(self.input_stack.pop())
        if self.output_stack:
            return self.output_stack[-1]   # Peek at the top of output stack
        else:
            return None   # Queue is empty

    def is_empty(self):
        """Check if the queue is empty."""
        return not self.input_stack and not self.output_stack

    def size(self):
        """Return the number of elements in the queue."""
        return len(self.input_stack) + len(self.output_stack)
```

...

**PULSE response:**

```python
class QueueUsingStacks:
    def __init__(self):
        self.input_stack = []
        self.output_stack = []

    def enqueue(self, data):
        """
        Adds an element to the rear of the queue.

        Args:
            data: The element to be added.
        """
        self.input_stack.append(data)

    def dequeue(self):
        """
        Removes and returns the element at the front of the queue.

        Returns:
            The element at the front of the queue.
            Raises an IndexError if the queue is empty.
        """
        if not self.output_stack:
            while self.input_stack:
                self.output_stack.append(self.input_stack.pop())
        if not self.output_stack:
            raise IndexError("Cannot dequeue from an empty queue")
        return self.output_stack.pop()

    def peek(self):
        """
        Returns the element at the front of the queue without removing it.

        Returns:
            The element at the front of the queue.
            Raises an IndexError if the queue is empty.
        """
        if not self.output_stack:
            while self.input_stack:
                self.output_stack.append(self.input_stack.pop())
        if not self.output_stack:
            raise IndexError("Cannot peek into an empty queue")
        return self.output_stack[-1]

    def is_empty(self):
        """
        Checks if the queue is empty.

        Returns:
            True if the queue is empty, False otherwise.
        """
        return not self.input_stack and not self.output_stack

    def size(self):
        """
        Returns the number of elements in the queue.

        Returns:
            The number of elements in the queue.
        """
        return len(self.input_stack) + len(self.output_stack) ...
```

