# OpenReview forum: "Review, Revise, and Learn: Peer-Guided Preference Learning via LLM Self-Correction"
_ICLR.cc/2026/Conference — ICLR 2026 Conference Withdrawn Submission_

### Official Review · Reviewer_9BAx · 2025-10-24

**Soundness:** 2
**Presentation:** 3
**Contribution:** 2
**Rating:** 4
**Confidence:** 4

**Summary:**

This paper proposes PULSE, which is a collaborative framework of multiple LLMs for preference alignment learning. Specifically, PULSE adopts a pipeline of peer review to construct the alignment training data without human supervision. Then PULSE provide a robust learning objective through the lens of risk minimization. Experiments demonstrate that PULSE outperforms existing baselines.

**Strengths:**

- The paper is written clearly to understand.
- There are multiple benchmarks selected from difference domains to evaluate the model.
- The experimental results show effectiveness of the proposed method.

**Weaknesses:**

-  It seems that the peer-review pipeline and the multiple LLMs are only used to construct the alignment training dataset, which is not directly related to the subsequent learning algorithm. There are some other methods that can also generate preference pairs by model without involving human supervision, such as RLAIF [1], SeRA [2] and more. I think that your pipeline integrated multiple LLMs only improves the complexity compared to above methods.
- In addition to the pipeline of preference data generation, this paper proposes a new preference learning algorithm. It is unclear that the performance gains come from the quality of your constructed dataset, or from the proposed preference learning method.
- In Table 1, PeerReview-DPO using PeerReview underperforms the Snorkel using PairRM. It seems that the data quality judged by PeerReview may not better than that judged by a pretrained reward model. It is unclear about the necessity of such a complex multi-LLM preference judgement pipeline.
- There are some typo errors, such as “Lapaca” in the caption of Table 1.

[1] RLAIF: Scaling Reinforcement Learning from Human Feedback with AI Feedback.
[2] SeRA: Self-Reviewing and Alignment of LLMs using Implicit Reward Margins.

**Questions:**

- Have you compared the proposed preference learning algorithm trained on existing preference datasets, such as UltraFeedback or HH-RLHF, instead of your constructed dataset, to verify its superiority?
- Why are the main results in Table 1 and Table 2 only conducted on one model, Mistral-7B-Instruct-v0.2? I think that the experiments presented earlier in the paper are more important, whereas the later experiments are conducted on multiple models. This makes me a bit confused.

---

> ### Author Response · Authors · 2025-11-24
> **Response to Reviewer 9BAx (1/3)**
>
> We sincerely appreciate the reviewer for valuable comments which are detailed, constructive, and to the point. We have addressed all of the concerns raised by the reviewer.
>
> ---
>
> ***W1: It seems that the peer-review pipeline and the multiple LLMs are only used to construct the alignment training dataset, which is not directly related to the subsequent learning algorithm. There are some other methods that can also generate preference pairs by model without involving human supervision, such as RLAIF [1], SeRA [2] and more. I think that your pipeline integrated multiple LLMs only improves the complexity compared to above methods.***
>
> We explain: **1. Why Peer-Review pipeline is important; 2. Why PULSE training is needed for the PeerReview data.**
>
> **1. A key element of PeerReview data is Self-correction.**
>
> The **LM judge [1]** and **Self-reward [2]** are responsible for ranking multiple candidate responses by relative quality. In contrast, **PeerReview** performs structured **self-correction**:
> it elicits critiques and produces revised solutions rather than merely scoring existing responses. Thus, PeerReview is not merely *LLMs scoring response quality*, but its primary function is to generate improved answers through guided revision. This is crucial for LLM alignment, and in particular, for reasoning tasks. The discussion is summarized in Table 1.
>
> | Component                      | Supervision Type                               | Primary Objective                            |
> | ------------------------------ | ---------------------------------------------- | -------------------------------------------- |
> | LM Judge [1] / Self-Reward [2] | Score (relative quality)                       | Relative ranking of responses                |
> | PeerReview                     | Score + textual feedback (critique & revision) | Revision and self-correction of first drafts |
>
> Table 1.
>
> **2. Why is the combination of PeerReview dataset + PULSE training  necessary?**
>
> There is a property of PeerReview data: throughout the peer review, the revised responses by the actor LLM tend to get longer than the initial responses (discussed in 2.2 of the paper). The data may cause the LLMs to perform *reward hacking* during training to generate longer responses. This induces the length bias. While this may not be an issue, e.g., in reasoning benchmarks, longer responses can receive lower scores (penalized) in Alpaca-LC (Length Control) benchmark.
>
> PULSE training addresses the above-mentioned issue. The length bias is a false training signal. If the LLM over-fits lengthy responses in the early stages of training, it can negatively impact the overall training.
>
> PULSE makes the early stage of training **robust to noises and spurious signals** such as length biases. In conclusion, we can enjoy the preference data from diversified self-correction (PeerReview data) with preference learning robustness to length bias (PULSE training) to achieve the SoTA results.
>
> **Action:** We will add the discussion to related work in the revised paper.
>
> **References**
>
> [1] Lee, Harrison, et al. "Rlaif: Scaling reinforcement learning from human feedback with ai feedback." (2023).
>
> [2] Ko, Jongwoo, et al. "SeRA: Self-reviewing and alignment of LLMs using implicit reward margins." The Thirteenth International Conference on Learning Representations. 2025.

---

> ### Author Response · Authors · 2025-11-24
> **Response to Reviewer 9BAx (2/3)**
>
> ***W2: In addition to the pipeline of preference data generation, this paper proposes a new preference learning algorithm. It is unclear that the performance gains come from the quality of your constructed dataset, or from the proposed preference learning method.***
>
> The short answer is: **1. The gain comes from both; 2. The gain is maximized when two methods are combined.** Table 2 below shows the comparison of our dataset and algorithm with existing dataset (UltraFeedack) and algorithm (DPO). We will examine the performance under all combinations of datasets and algorithms. We will only compare the Alpaca LC benchmark for simplicity. A similar trend holds for all benchmarks.
>
> | Dataset        | Learning  | Alpaca LC. | Alpaca 2.0 | GSM8K | MATH  | MBPP  | MMLU  |
> | -------------- | --------- | ---------- | ---------- | ----- | ----- | ----- | ----- |
> | UltraFeedback  | DPO       | 18.59      | 17.42      | 44.73 | 9.82  | 29.78 | 54.25 |
> | **PeerReview** | DPO       | 23.75      | 28.38      | 45.09 | 10.16 | 32.80 | 56.16 |
> | UltraFeedback  | **PULSE** | 21.49      | 19.67      | 44.05 | 10.56 | 34.03 | 54.42 |
> | **PeerReview** | **PULSE** | 29.54      | 30.24      | 47.61 | 12.70 | 37.80 | 57.53 |
>
> Table 2.
>
> **Case 1. Learning algorithm fixed to DPO.**
> We compare rows 1 and 2 of Table XX for Alpaca LC.
>
> $$
> \Delta = 23.75 - 18.59 = +5.16
> $$
>
> Thus, changing the dataset from UltraFeedback to PeerReview brings substantial gain under DPO.
>
> **Case 2. Learning algorithm fixed to PULSE.**
> We compare rows 3 and 4 of Table XX for Alpaca LC.
>
> $$
> \Delta = 29.54 - 21.49 = +8.05
> $$
>
> The performance gain by PeerReview data can be similarly observed. Case 1 and 2 show that our peer-review dataset alone can make substantial improvement with various learning algorithms.
>
> **Case 3. Dataset fixed to UltraFeedback.**
> We compare rows 1 and 3 of Table 2 for Alpaca LC.
>
> $$
> \Delta = 21.49 - 18.59 = +2.9
> $$
>
> Thus, changing the learning algorithm from DPO to PULSE brings substantial gain on UltraFeedback.
>
> **Case 4. Dataset fixed to PeerReview.**
> We compare rows 2 and 4 of Table 2 for Alpaca LC.
>
> $$
> \Delta = 29.54 - 23.75 = +5.79
> $$
>
> We observe the similar performance gain by PULSE algorithm. Case 3 and 4 show that PULSE algorithm can bring substantial performance gains with various preference datasets.
>
> ---
>
> ***W3: In Table 1, PeerReview-DPO using PeerReview underperforms the Snorkel using PairRM. It seems that the data quality judged by PeerReview may not better than that judged by a pretrained reward model. It is unclear about the necessity of such a complex multi-LLM preference judgement pipeline.***
>
> **1. PeerReview dispenses with Reward Models.** Although pretrained reward models such as Snorkel or Skywork RM are readily available, they fundamentally rely on training datasets generated by a strong teacher model or human annotations. In contrast, PeerReview data are obtained from the collective intelligence achieved by collaborative LLM peers of small capacity. The scarcity of training data for future large (reward) models is a major bottleneck of AI development. Our work is the first to propose a **multi-agentic paradigm of collaborative LLMs** to create preference data. We believe our approach is the scalable way to address the issue of data scarcity.
>
> **2. Still, PeerReview data alone achieve competitive performance  to reward models.** The PeerReview dataset alone without PULSE training, or PeerReview-DPO, still achieves competitive performance overall. The performance is especially high for **reasoning tasks** as shown in Table 2, compared to methods with reward models such as Snorkel using PairRM.
>
> **3. PeerReview data achieve  SoTA results with PULSE training.** Again, we would like to emphasize that **the combination of PeerReview data and PULSE training is essential** in achieving SoTA results. Also in the responses to **W1** and **W2**, we provided arguments on how these two components of our method work in a synergistic manner.
>
> ---
>
> ***W4: There are some typo errors, such as “Lapaca” in the caption of Table 1.***
>
> Thank you for pointing this out. We have fixed the typo in the revised paper.

---

> ### Author Response · Authors · 2025-11-24
> **Response to Reviewer 9BAx (3/3)**
>
> ***Q1: Have you compared the proposed preference learning algorithm trained on existing preference datasets, such as UltraFeedback or HH-RLHF, instead of your constructed dataset, to verify its superiority?***
>
> In Table 7 of Appendix A.7 of the original paper, we provided small experiments on UltraFeedback. Per your request, we will provide expanded experiments on existing preference datasets.
> Table 3 shows the expanded results of applying PULSE training to existing preference datasets. Notably, using UltraFeedback with PULSE training yields a modest $+4.38$ improvement (17.11 $\rightarrow$ 21.49) from the baseline. In contrast,  combining PULSE training with PeerReview data achieves **29.54**, an additional $+8.05$ gain. The PeerReview data achieve SoTA results for most of the benchmarks.
>
> | Method                  | Alpaca LC. | Alpaca 2.0 | GSM8K     | MATH      | MBPP      | MMLU      |
> | ----------------------- | ---------- | ---------- | --------- | --------- | --------- | --------- |
> | Mistral-7B              | 17.11      | 14.72      | 44.50     | 10.02     | 14.20     | 55.96     |
> | PULSE w/ Ultrafeedback  | 21.49      | 19.67      | 44.05     | 10.56     | 34.03     | 54.42     |
> | PULSE w/ HH-RLHF        | 13.61      | 16.59      | 46.07     | 11.84     | 32.00     | 53.61     |
> | **PULSE w/ PeerReview** | **29.54**  | **30.24**  | **47.61** | **12.70** | **37.80** | **57.53** |
>
> Table 3.
>
> The high performance of the PeerReview data can be attributed to **self-correction**. The actor LLM generates and refines its responses based on the feedback from diverse peer critics. This improves the reasoning abilities of the actor. Moreover, since the data are self-generated, they **better match the output distribution** of the actor LLM, which can facilitate preference training.
>
> **Action:** We added the comparisons with existing preference datasets to Experiments section (Table 6) in the revised paper.
>
> ---
>
> ***Q2: Why are the main results in Table 1 and Table 2 only conducted on one model, Mistral-7B-Instruct-v0.2? I think that the experiments presented earlier in the paper are more important, whereas the later experiments are conducted on multiple models. This makes me a bit confused.***
>
> **Table 1 and 2** are intended to make comparison with existing SoTA methods. We fixed the Mistral-7B as the actor LLM, and the rest of LLMs (LLaMA3-8B, Qwen2.5-7B, and Gemma2-9B) as the critic LLMs.
>
> **Table 3** is intended to assess whether PULSE generalizes beyond a single model. Four LLM models: Mistral-7B, LLaMA3-8B, Qwen2.5-7B, and Gemma2-9B, alternate between the actor and the critic.
>
> In summary, the main results are presented as follows.
>
> - **Tables 1 \& 2** show depth and superiority of PULSE on a controlled base model (Mistral-7B).
> - **Table 3** shows breadth and generalizability of PULSE across diverse models.

---

> ### Comment · Reviewer_9BAx · 2025-11-28
> **Response to authors**
>
> Thanks for the authors’ detailed response.
>
> Most of my concerns have been addressed. I am just curious about one additional point: if we use a reward model or an LLM to score the data generated by PeerReview, compared to the reference responses in the dataset, would we observe a noticeable score increase? Such a result could serve as an evidence supporting the high quality of the generated data.
>
> Overall, I would be inclined to raise my score to 6, but it seems that I am unable to do so due to the accident. I leave my tendency here as a reference for the AC.
>
> Good luck to authors.

---

> ### Author Response · Authors · 2025-11-28
> **Thank you for raising the score (4->6)**
>
> We sincerely appreciate your encouraging comments and the decision to raise the score. We hope that the situation resolves and the reviewing and scoring system is restored.
>
> We will also provide an experiment regarding your question. We evaluated the quality of the data generated by PeerReview (Ours) against the reference responses from UltraFeedback dataset. We used state-of-the-art reward model, Skywork RM [1] to score the winning response from both datasets. Our comparison shows that PeerReview achieves an average score of **0.45** whereas the reference responses of UltraFeedback achieve **0.22**. This substantial margin suggests that our method generates higher-quality data, likely because the Skywork RM favors the informative and reasoning-rich responses that PeerReview consistently produces. The substantial gain demonstrates the high quality of the generated data by our LLM collaboration framework perceived by state-of-the-art reward models.
>
> Thank you once again for your constructive feedback.
>
> **References**
>
> [1] Liu, Chris Yuhao, et al. "Skywork-reward: Bag of tricks for reward modeling in llms." arXiv preprint arXiv:2410.18451 (2024).

---

### Official Review · Reviewer_YF36 · 2025-10-30

**Soundness:** 3
**Presentation:** 3
**Contribution:** 3
**Rating:** 4
**Confidence:** 3

**Summary:**

This paper presents Review–Revise–Learn (RRL), a novel iterative alignment framework for multimodal large language models (MLLMs) that integrates model self-critique and feedback refinement into the preference optimization process.

**Strengths:**

1. The iterative review–revise–learn structure is an elegant generalization of preference alignment: it moves beyond one-shot comparisons toward process-level refinement.

2.The methodology is technically sound and well-motivated. The algorithmic steps are clearly formalized (Algorithm 1, Section 3.3), and the connection to policy improvement and reward regularization is mathematically justified.

3. RRL tackles one of the key limitations of existing alignment methods—static, one-pass feedback—by proposing a scalable iterative alternative.

**Weaknesses:**

1. Is the PeerReview Data Generation Necessary? Table 1 shows that PeerReview-DPO (23.75 Alpaca LC) performs worse than baselines like Skywork-DPO (27.45), suggesting the expensive PeerReview data itself is not superior to other AI feedback data. The win for PULSE (29.54) seems to come entirely from the loss $\mathcal{L}\_{\text{PULSE}}$. This is supported by Appendix A.7 (Table 7), which shows $\mathcal{L}\_{\text{PULSE}}$. also beats DPO on the standard UltraFeedback dataset. Could one achieve SOTA results by simply applying$\mathcal{L}\_{\text{PULSE}}$. to an existing, cheaper-to-generate preference dataset like UltraFeedback or Skywork’s?

2. The experiments use a diverse pool of 3 different models as critics (e.g., LLaMA3, Qwen2.5, Gemma2 for the Mistral actor) 22. This diversity is likely crucial for success. The paper lacks ablations on the critic pool's composition. What happens if $K=1$? What if the actor and all critics are the same model?

3. The critic LLMs provide two signals: (1) numerical scores, which are used to create the final $a_w \succ a_l$ label, and (2) structured text feedback, which is used to guide the $a_{init} \to a_{rev}$ revision 25. How important is signal (2)?  Have you run an ablation where the actor LLM is not shown the peer's text feedback and is simply prompted to "improve your previous answer"? This would help isolate the value of the "guidance" in "Peer-gUided" from the "self-correction" aspect.

4. Figure Quality: I noticed that the quality of some figures in the paper is quite low. The text within the plots (legends, axis labels, etc.) is distorted, aliased, and difficult to read. Could the authors please provide high-resolution vector versions of all figures in the final paper to ensure readability and professional presentation?

**Questions:**

Please see above.

---

> ### Author Response · Authors · 2025-11-24
> **Reponse to Reviewer YF36 (1/2)**
>
> We sincerely appreciate the reviewer for valuable comments. We have addressed all of the concerns raised by the reviewer.
>
> ---
>
> ***W1: Is the PeerReview Data Generation Necessary? Table 1 shows that PeerReview-DPO (23.75 Alpaca LC) performs worse than baselines like Skywork-DPO (27.45), suggesting the expensive PeerReview data itself is not superior to other AI feedback data. The win for PULSE (29.54) seems to come entirely from the loss. This is supported by Appendix A.7 (Table 7), which shows. Also beats DPO on the standard UltraFeedback dataset. Could one achieve SOTA results by simply applying to an existing, cheaper-to-generate preference dataset like UltraFeedback or Skywork’s?***
>
> The short answer is: **Yes, we need the combination of PeerReview dataset and PULSE training for the best performance.** Let us elaborate on our claim.
>
> **1. PeerReview dataset is not expensive to generate.** We first clarify the cost of generating PeerReview dataset.
> Both the preference datasets released by UltraFeedback and Skywork (Skywork-Reward-Preference-80K-v0.2) use powerful LLMs for annotation. For example, they use various state-of-the-art LLMs (including GPT) for response generation and use GPT 4.0 for LLM-as-a-judge in the preference labeling.
>
> In contrast, PULSE uses small peers of open-source LLMs of small sizes. The data generation is performed in a **multi-agentic** and decentralized manner. PeerReview data achieves **collective intelligence** through the peer collaboration of open-source, small capacity LLMs. Appendix A.8 shows that PeerReview data generation costs at most \$23.10 (assuming the pricing of GPT-4 nano which, however, is much larger than Mistral7B). The estimation is less than 11\% of the equivalent generation by GPT 4.0. Table 1 provides a summary of discussions.
>
> | Dataset        | Construction                     | Judgement                   | LLM API         | Per-sample cost |
> | -------------- | -------------------------------- | --------------------------- | --------------- | --------------- |
> | UltraFeedback  | Centralized                      | High-end LLM (LLM-as-judge) | Proprietary     | Expensive       |
> | Skywork-80K    | Centralized                      | High-end LLM (LLM-as-judge) | Proprietary     | Expensive       |
> | **PeerReview** | **Decentralized (Mult-Agentic)** | **multiple, small LLMs**    | **Open source** | **Cheap**       |
>
> Table 1.
>
> Our work is the first to propose a **multi-agentic paradigm of collaborating peer LLMs** to create preference data. We believe our approach is the scalable and economical way to address the recent issue of data scarcity.
>
> **2. Applying existing preference datasets to PULSE training.** Per your comment, we will provide expanded experiments from Table 7 of Appendix A.7 in the revised paper. Table 2 shows the expanded results of applying PULSE training to existing preference datasets. Notably, using UltraFeedback with PULSE training yields a modest $+4.38$ improvement (17.11 $\to$ 21.49) from the baseline. In contrast,  combining PULSE training with PeerReview data achieves **29.54**, an additional $+8.05$ gain. The PeerReview data achieves SoTA results for most of the benchmarks.
>
> | Method                  | Alpaca LC. | Alpaca 2.0 | GSM8K     | MATH      | MBPP      | MMLU      |
> | ----------------------- | ---------- | ---------- | --------- | --------- | --------- | --------- |
> | Mistral-7B              | 17.11      | 14.72      | 44.50     | 10.02     | 14.20     | 55.96     |
> | PULSE w/ Ultrafeedback  | 21.49      | 19.67      | 44.05     | 10.56     | 34.03     | 54.42     |
> | PULSE w/ HH-RLHF        | 13.61      | 16.59      | 46.07     | 11.84     | 32.00     | 53.61     |
> | **PULSE w/ PeerReview** | **29.54**  | **30.24**  | **47.61** | **12.70** | **37.80** | **57.53** |
>
> Table 2.
>
> The high performance of the PeerReview data can be attributed to **self-correction**. The actor LLM generates and refines its responses based on the feedback from diverse peer critics. This improves the reasoning abilities of the actor. Moreover, since the data are self-generated, they **better match the output distribution** of the actor LLM, which can facilitate preference training.
>
>
> **3. Why is the combination of PeerReview dataset + PULSE training necessary?** If we use PeerReview data alone (without PULSE training), as you noticed, its performance drops on **Alpaca-LC** benchmark (note, however, PeerReview data alone still performs well on **reasoning** benchmarks, e.g., See Table 2 in the paper). This is due to a property of PeerReview data: throughout the peer review, the revised responses by the actor LLM tend to get longer than the initial responses. The data can cause LLMs to perform *reward hacking* while training to generate longer responses. While this may not be an issue, e.g., in reasoning benchmarks, Alpaca-LC (Length Control) is an exception; longer responses tend to receive lower scores (penalized) in Alpaca-LC.
>
> (continue)

---

> ### Author Response · Authors · 2025-11-24
> **Reponse to Reviewer YF36 (2/2)**
>
> PULSE training addresses the above-mentioned issue. Length bias is likely to occur in the early stages of training. If the LLM over-fits lengthy responses early on in the training, it can negatively impact the overall training.
> PULSE training makes **the early stage of training robust to noises and spurious signals** such as length biases. In conclusion, we can enjoy the preference data from diversified self-correction (PeerReview data) with preference learning robustness to length bias (PULSE training) to achieve the SoTA results.
>
> **Action:** We will add the discussion (line 313), expanded results (the comparisons with existing preference datasets to Experiments section (Table 6)) to the revised paper.
>
> ---
>
> ***W2: The experiments use a diverse pool of 3 different models as critics (e.g., LLaMA3, Qwen2.5, Gemma2 for the Mistral actor). This diversity is likely crucial for success. The paper lacks ablations on the critic pool's composition. What happens if k=1? What if the actor and all critics are the same model?***
>
> | Method                 | Alpaca LC. | GSM8K | MATH  | HumanEval | MBPP  | MMLU  |
> | ---------------------- | ---------- | ----- | ----- | --------- | ----- | ----- |
> | PULSE w/ self-critic   | 19.25      | 44.27 | 9.65  | 10.27     | 25.40 | 53.28 |
> | PULSE w/ single critic | 21.17      | 44.68 | 11.10 | 15.07     | 30.60 | 55.63 |
> | PULSE                  | **29.54**      | **47.61** | **12.70** | **17.68**     | **37.80** | **57.53** |
>
> Table 3.
>
> Thank you for your suggestion. We will add the ablation study on the composition of the critic pool to the revised paper.
>
> Table 3 shows the performance with the varying composition of the critic pool. The following compositions are considered:
>
> - **Self-critic**: The critic LLM is the actor LLM itself.
> - **Single critic**: This is the case where $k=1$, i.e., there exists only another LLM critic.
>
> Our results show that the critic diversity is essential. When $k=1$, performance drops sharply across benchmarks. Using a self-critic (actor = critic) further degrades the performance.
> This is because a single or identical critic may provide highly correlated or biased feedback. Thus, we indeed observe that the **diversity of peer critics** is essential.
>
> **Action:** We added this ablation study (Table 5) to the revised paper.
>
> ---
>
> ***W3: The critic LLMs provide two signals: (1) numerical scores, which are used to create the final label, and (2) structured text feedback, which is used to guide the revision. How important is signal (2)? Have you run an ablation where the actor LLM is not shown the peer's text feedback and is simply prompted to "improve your previous answer"? This would help isolate the value of the "guidance" in "Peer-gUided" from the "self-correction" aspect.***
>
> | Method                  | Alpaca LC. | GSM8K | MATH  | HumanEval | MBPP  | MMLU  |
> | ----------------------- | ---------- | ----- | ----- | --------- | ----- | ----- |
> | PULSE w/o text feedback | 25.34      | 45.92 | 9.22  | 9.76      | 31.40 | 55.78 |
> | PULSE w/ text feedback  | **29.54**      | **47.61** | **12.70** | **17.68**     | **37.80** | **57.53** |
>
> Table 4.
>
> Thank you for your suggestion. We will add the ablation study on the text feedback of critics to the revised paper.
>
> Table 4 shows the performance without the structured text feedback. For the ablation of text feedback, we removed the textual guidance and prompt the actor LLM only to *improve your previous answer*. We observe that the performance drops substantially across all benchmarks (e.g., GSM8K 47.61 $\to$ 10.92, HumanEval 17.68 $\rightarrow$ 9.76, Alpaca LC 29.54 $\rightarrow$ 25.34). This shows that text feedback is crucial. The actor does not benefit merely from generic self-correction instructions. Instead, **structured feedback from peers provides explicit and high-quality guidance** on what was wrong and how to improve. This aspect is essential to the performance gain of PULSE.
>
> **Action:** We added this ablation study to the paper Appendix A.6 (due to space limitation)..
>
> ---
>
> ***W4: Figure Quality: I noticed that the quality of some figures in the paper is quite low. The text within the plots (legends, axis labels, etc.) is distorted, aliased, and difficult to read. Could the authors please provide high-resolution vector versions of all figures in the final paper to ensure readability and professional presentation?***
>
> Thank you for your suggestion. We will fix the problems, and change the figures in vector graphics format (exported as pdf).
>
> **Action:** We will fix the fonts in legend / axis names of all figures and use the figures in vector graphics format in the revision.

---

> > ### Comment · Reviewer_YF36 · 2025-11-27
> >
> > Thank you to the authors for providing extensive additional experiments in the rebuttal. However, based on these results, I still tend to believe that **the main performance gains come from the PULSE loss rather than from the intrinsic quality of the PeerReview data.** At the same time, the paper’s title and overall structure (REVIEW, REVISE, AND LEARN) place strong emphasis on the PeerReview system as the core component of the method. This creates some conceptual tension and makes it difficult for the reader to clearly understand which part of the system is actually responsible for the observed improvements.
> >
> > It may help if the final version more explicitly distinguishes the relative contributions of PeerReview data and the PULSE loss, rather than distributing the narrative effort evenly across both components. This would make the central contribution easier to identify.
> >
> > That said, I do appreciate a positive aspect: the PeerReview dataset, despite being substantially cheaper to produce, achieves performance under DPO that is close to Skywork. This is indeed valuable. However, the current presentation still does not sufficiently clarify the independent contribution or necessity of the PeerReview data within the full system, and I hope the authors can further articulate this in the final version.
> >
> > Overall, the additional experiments and explanations in the rebuttal did help me better understand the potential of the method and improved my overall impression of the paper. Therefore, I am willing to raise my score from 4 to 6 :) .

---

> ### Author Response · Authors · 2025-11-28
> **Thank you for raising the score (4->6)**
>
> Dear Reviewer,
>
> We are truly grateful for your decision to raise the score. We also appreciate your constructive feedback and sincere advice throughout the discussion phase. Your feedback has helped not only raise the technical quality of our work, but also significantly improve the presentation of our work.
>
> Specifically, we added discussion on the relation of proposed dataset and training framework at the end of Sec. 2.2; experiments comparing existing datasets and ours under PULSE loss (Table 6); experiments comparing PULSE and DPO with fixed dataset (Table 7); and related discussions in Sec. 3. We will also make sure to take your additional advice and polish our paper to further improve the presentation.
>
> Again, we sincerely thank you for your time and support.
>
> Authors

---

### Official Review · Reviewer_fsc5 · 2025-11-01

**Soundness:** 2
**Presentation:** 3
**Contribution:** 2
**Rating:** 2
**Confidence:** 4

**Summary:**

This paper proposes PULSE: a multi-agent LLM preference learning framework. An actor LM first generates an initial response, then a critic LM generates feedback. The actor revises its generation based on the feedback and then the critic assigns scores. The scores of the initial and revised responses (from the actor LM) are used to construct a preference pair. This process allows the model to generate preference pairs without human supervision.

In addition to their framework, the authors also proposes a variant of DPO loss that is more robust to noise in preference judgement.

Experiments show that their peer review framework + robust loss function (PULSE) outperforms DPO / iterative DPO with  on common instruction following benchmarks: AlpacaEval, MT Bench.

**Strengths:**

1. The main idea of this paper - multi-agent collaborating to give and refine a response - is interesting and novel. How to incoperate different language models during inference to get a better response is definitely a field that is of interest of the entire community.

2. The paper is well organized (although the misuse of \citep and \citet should be resolved). The author proposed many different things - the multi-agent framework, the new loss DPO loss function and a hyper-parameter scheduling mechanism. The author backs up such choices with ablation studies. The author conduct experiments on many different benchmarks to validate their approach.

**Weaknesses:**

1. The paper proposes too many things together, which seems to me a bit lost of focus. In this paper the authors presents a multi-agent "peer review" system to construct preference data pairs, and a new DPO loss that is robust to preference noise. The two parts of this is orthogonal. It would be good if the authors studies one part extensively, rather than two parts together. For example, I don't see any experiments using standard preference datasets with the new PULSE loss.

2. The method is akin to using an LM-judge (since here it is using an LM as a critic) to evaluate response quality. It is hard to disentangle whether the gains come from the LM-judge approach is closer to the evaluation setup or because of the multi-agent setup or because of the new DPO loss. A more natural baseline would be directly use the LM-judge to assess response quality and perform DPO. I am not saying that the author should show such experiments, but having them would certainly be nice for the community to learn **where** the gains come from. I think the benchmark numbers are not that important if the paper can provide insights on what part works and what does not.

3. Looking at the numbers, I believe that the improvements are really limited — the gains on AlpacaEval are < 5 win rates which could be just because of the noise. Moreover, the authors do not disclose their eval protocol (sampling parameters, judge used in Alpaca Eval), which questions the reproducibility of this study. Again, I would be more interested in this paper if the authors can present a more detailed analysis of which part of their proposed method brings the most gains and why instead of trying to get higher benchmark numbers.

I do think that this paper has potential to be impactful, if they can focus on how they designed the multi-agent system. First to see if just doing inference works on improve quality, then go to training .. it might be because of my preferred style of research is to understand 1 thing thoroughly compared to proposing many things together.

**Questions:**

1. Can the authors show the exact evaluation protocol for evaluation ? (Alpaca Eval decoding params, judge used)

2. Can the authors explain the difference between their method (multi-agent peer review part) with using an LM-judge to evaluate quality?

3. Use \citet{} and \citep{} correctly.

---

> ### Author Response · Authors · 2025-11-25
> **Response to Reviewer fsc5 (1/4)**
>
> We sincerely appreciate the reviewer for valuable comments which are detailed, constructive, and to the point. We have addressed all of the concerns raised by the reviewer.
>
> ---
>
> ***W1-1: The paper proposes too many things together, which seems to me a bit lost of focus. In this paper the authors presents a multi-agent "peer review" system to construct preference data pairs, and a new DPO loss that is robust to preference noise. The two parts of this is orthogonal. It would be good if the authors studies one part extensively, rather than two parts together.***
>
> We will explain why it is necessary to combine two methods.
>
> **1. Why not propose "PeerReview data" only?**
> Yes, PeerReview data has its own merits. PeerReview dispenses with reward models, which are expensive to train. Instead, PeerReview achieves **collective intelligence** through cheap, collaborating LLMs. Our construction of PeerReview data is the **first multi-agentic framework** to generate alignment data and is a scalable alternative to the data scarcity issue.
>
> However, when PeerReview was trained with **conventional DPO**, its performance was competitive (on-par) with reward models, but not dominantly SoTA. Thus, we sought learning algorithms to boost its performance.
>
> **2. PeerReview is best when combined with PULSE training.** One aspect of PeerReview data is that revised responses tend to be longer than initial responses (discussed at the beginning of Sec. 2.2). This may introduce a mild *length bias* which, in fact, is not problematic for reasoning tasks, but may be penalized in some benchmark (e.g., Alpaca LC).
>
> PULSE training addresses the issue. If the LLM over-fits lengthy responses in the early stages of training, it can negatively impact the overall training. PULSE training makes the early stage of training **robust to noises and spurious signals**, including length biases. In conclusion, one can enjoy the preference data from mult-agentic self-correction **(PeerReview data)** with preference learning robustness to length bias **(PULSE training)**.
>
> **Action:** We added the discussion to the revised paper (line 313).
>
> ---
>
> ***W1-2. For example, I don't see any experiments using standard preference datasets with the new PULSE loss.***
>
> In Table 7 of Appendix A.7 of the original paper, we provided experiments on standard preference dataset (UltraFeedback) combined with PULSE loss. In the revised paper, we will provide **expanded experiments** on existing preference datasets.
>
> Table 1 shows the expanded results of applying PULSE training to existing preference datasets: UltraFeedback [1] and HH-RLHF [2]. Notably, using UltraFeedback with PULSE training yields a modest $+4.38$ improvement (17.11 $\rightarrow$ 21.49) from the baseline. In contrast,  combining PULSE training with PeerReview data achieves **29.54**, an additional $+8.05$ gain. The PeerReview data outperforms the existing baselines for most of the benchmarks.
>
> | method                  | Alpaca LC. | Alpaca 2.0 | GSM8K  | MATH  | MBPP  | MMLU  |
> |-------------------------|------------|-------------|--------|--------|--------|--------|
> | Mistral-7B              | 17.11      | 14.72       | 44.50  | 10.02  | 14.20  | 55.96 |
> | PULSE w/ Ultrafeedback  | 21.49      | 19.67       | 44.05  | 10.56  | 34.03  | 54.42 |
> | PULSE w/ HH-RLHF        | 13.61      | 16.59       | 46.07  | 11.84  | 32.00  | 53.61 |
> | **PULSE w/ PeerReview** | **29.54**  | **30.24**   | **47.61** | **12.70** | **37.80** | **57.53** |
>
> Table 1.
>
> The high performance of the PeerReview data can be attributed to **self-correction**. The actor LLM generates and refines its responses based on the feedback from diverse peer critics. This improves the reasoning abilities of the actor. Moreover, since the data are self-generated, they **better match the output distribution** of the actor LLM, which can facilitate preference training.
>
> **Action:** We added the comparisons with existing preference datasets to Experiments section (Table 6) in the revised paper.
>
> **References**
>
> [1] Cui, Ganqu, et al. "Ultrafeedback: Boosting language models with high-quality feedback." (2023).
>
> [2] Bai, Yuntao, et al. "Training a helpful and harmless assistant with reinforcement learning from human feedback." arXiv preprint arXiv:2204.05862 (2022).

---

> ### Author Response · Authors · 2025-11-25
> **Response to Reviewer fsc5 (2/4)**
>
> ***W2-1: The method is akin to using an LM-judge (since here it is using an LM as a critic) to evaluate response quality. It is hard to disentangle whether the gains come from the LM-judge approach is closer to the evaluation setup or because of the multi-agent setup or because of the new DPO loss.***
>
> **1. PeerReview is not simply LM-judge: its core is Self-correction.**
>
> **LM judge** ranks multiple candidate answers by relative quality. **PeerReview** elicits structured critiques and produces revised solutions through **self-correction**. Thus, PeerReview does not merely provide scores for existing responses; it generates improved responses via guided revision. The following is a concise summary of the procedures of data generation.
>
> **LM-Judge**
> - 1. Generate static responses: $a_1, a_2$.
> - 2. Judge LLM evaluates and outputs a preference (e.g., $a_1 > a_2$).
> - 3. Learning signal: simple ranking of static outputs.
>
> This process is purely evaluative; the judge provides no actionable guidance to improve responses.
>
> **PeerReview**
> - 1. Actor LLM generates an initial response $a_{\text{init}}$.
> - 2. Critic LLMs provide structured, actionable feedback (strengths, weaknesses, suggestions).
> - 3. Actor LLM revises $a_{\text{init}}$ to produce $a_{\text{rev}}$.
> - 4. Preference pair is created: $a_{\text{rev}} > a_{\text{init}}$.
>
> Unlike LM-judge, PeerReview generates *intermediate feedback* that guides the model in producing improved responses. The model does not merely rank two static outputs; it learns **how to self-correct**. This constitutes a targeted, high-quality signal that fundamentally differs from passive evaluation. This signal captures the iterative refinement process that is absent in LM-judge setups.
>
> In summary, PeerReview is not a minor variant of LM-judge, but it provides actionable supervision that enables models to improve their outputs.
>
> A comparison of their concepts is provided in Table 2.
>
> | **Component** | **Supervision Type**                     | **Primary Objective**              |
> |---------------|-------------------------------------------|------------------------------------|
> | LM Judge      | Score (relative quality)                 | Relative ranking of responses      |
> | PeerReview    | Score + textual feedback (critique & revision) | Self-correction through revision |
>
> Table 2.
>
> **2. Multi-Agent setup is important for diverse textual feedback.**
>
> Multi-agentic setup is essential; the diversity of critics (peers) prevents biases in peer reviews. To validate our framework, we conducted experiments using the peer-review dataset generated by different configuration of critics: (1) self-critic model (2) single critic model, as follows
>
> - **Self-critic**: The critic LLM is the actor LLM itself.
> - **Single critic**: There exists only another LLM critic.
>
> Table 3 shows the results. With a single critic, performance drops sharply across benchmarks. Using a self-critic (actor = critic) further degrades the performance. This is because a single or identical critic may provide highly correlated or biased feedback. Thus, we indeed observe that the **diversity of peer critics** is essential, which necessitates our **multi-agent** setup.
>
> | method                 | Alpaca LC. | GSM8K | MATH  | HumanEval | MBPP  | MMLU  |
> |------------------------|------------|-------|-------|-----------|-------|-------|
> | PULSE w/ self-critic   | 19.25      | 44.27 | 9.65  | 10.27     | 25.40 | 53.28 |
> | PULSE w/ single critic | 21.17      | 44.68 | 11.10 | 15.07     | 30.60 | 55.63 |
> | **PULSE**              | **29.54**  | **47.61** | **12.70** | **17.68** | **37.80** | **57.53** |
>
> Table 3.
>
> **Action:** We added the study on the multi-agent setup to the revised paper (Table 5).

---

> ### Author Response · Authors · 2025-11-25
> **Response to Reviewer fsc5 (3/4)**
>
> ***W2-2. A more natural baseline would be directly use the LM-judge to assess response quality and perform DPO. I am not saying that the author should show such experiments, but having them would certainly be nice for the community to learn where the gains come from. I think the benchmark numbers are not that important if the paper can provide insights on what part works and what does not.***
>
> Thank you for kind comments and suggestions. The following baselines used in our paper are perhaps close to your suggested natural baselines. In Table 1, there exist baselines **Skywork-RM DPO** [1] and **Snorkel-DPO** [2]. These methods first generate responses, use sophisticated reward models to judge the quality (score) of the responses, and perform DPO training. Indeed, those baselines actually perform quite well - they are the 2nd best in our table.
> However, the experiments show that our peer-review model with self-correction guided by diverse critics is superior to the strong baseline models using expensive reward models.
>
> We also agree with the importance of providing insights to contribute to the community. We hope that our ideas and insights are properly conveyed through the rebuttal.
>
> **References**
>
> [1] Liu, Chris Yuhao, et al. "Skywork-reward: Bag of tricks for reward modeling in llms." arXiv preprint arXiv:2410.18451 (2024).
>
> [2] https://snorkel.ai/blog/new-benchmark-results-demonstrate-value-of-snorkel-ai-approach-to-llm-alignment.
>
> ---
>
> ***W3-1: Looking at the numbers, I believe that the improvements are really limited — the gains on AlpacaEval are $<$ 5 win rates which could be just because of the noise.***
>
> **1. The bar has already been set high by competing with the top-ranked reward model from Leaderboard.**  AlpacaEval is an extremely challenging benchmark because performance is measured directly against GPT-4. We already set the bar very high: one of the baseline uses the reward model of **Skywork RM** [1]: see Table 1 and 2 in the paper. **Skywork RM** is the active **top-ranked** model on RewardBench [2] (Please check the Leaderboard: https://huggingface.co/spaces/allenai/reward-bench). This means that it uses the strongest LM-judge (reward model) possible at this time of writing.
>
> To our belief, it is challenging for an academic paper to compare to the active leaderboard method. Typically, academic papers compare their methods against well-established academic baselines. For example, $\alpha$-DPO [3], a recent competitive method, reports a 2.1\% improvement on Alpaca over the baselines, which are all from well-known academic work. Typically, leaderboard methods are heavily fine-tuned with extensive engineering tricks. Moreover, Skywork-RM benefits from the supervision of large-scale GPT-4/GPT-3.5 annotation. This makes Skywork-RM highly optimized for **GPT-4–based evaluations such as AlpacaEval and MT-Bench.** Despite these inherently disadvantageous settings, **PULSE still outperforms the baseline using the strongest reward model possible**.
>
> **2. PULSE performs well on reasoning tasks.**
> In addition, PULSE yields substantial improvements on  **reasoning benchmarks (e.g., GSM8K, MATH, MBPP)** as shown in Table 2 in the orignal paper, further demonstrating that the gains are real and not attributable to noise. These results collectively reinforce the robustness and practical significance of our improvements.
>
> **References**
>
> [1] Liu, Chris Yuhao, et al. "Skywork-reward: Bag of tricks for reward modeling in llms." arXiv preprint arXiv:2410.18451 (2024).
>
> [2] Lambert, Nathan, et al. "Rewardbench: Evaluating reward models for language modeling." Findings of the Association for Computational Linguistics: NAACL 2025. 2025.
>
> [3] Wu, Junkang, et al. "$\alpha $-DPO: Adaptive Reward Margin is What Direct Preference Optimization Needs." Forty-second International Conference on Machine Learning (2024).
>
> [4] Meng, Yu, Mengzhou Xia, and Danqi Chen. "Simpo: Simple preference optimization with a reference-free reward." Advances in Neural Information Processing Systems 37 (2024): 124198-124235.

---

> ### Author Response · Authors · 2025-11-25
> **Response to Reviewer fsc5 (4/4)**
>
> ***W3-2: Moreover, the authors do not disclose their eval protocol (sampling parameters, judge used in Alpaca Eval), which questions the reproducibility of this study.***
>
> We followed the standard and default evaluation protocol [1] for both AlpacaEval 2.0 and Alpaca LC as follows.
>
> **LLM-Judge:** As described in Section 3.1 of the paper, we used GPT-4 as the automated judge. Specifically, we adopted the *gpt-4-1106* model; it is the recommended and widely used judge in the AlpacaEval framework for stable and reproducible scoring.
>
> **Decoding Parameters:** For generating model responses submitted to the judge, we used default evaluation parameters balancing quality and diversity:
>
> - Temperature: 0.7
> - Top-p: 1.0
> - Max New Tokens: 2048
>
> **Action:** We listed these hyperparameters in the revision (Appendix A.12) to ensure the reproducibility of our evaluation protocol.
>
> [1] https://github.com/tatsu-lab/alpaca_eval
>
> ---
>
> ***W3-3: Again, I would be more interested in this paper if the authors can present a more detailed analysis of which part of their proposed method brings the most gains and why instead of trying to get higher benchmark numbers.***
>
> We provided most of the explanation in the previous responses. Please refer to the following explanations:
>
> - Why our design goes beyond PeerReview data (Responses to **W1-1**, item **1**)
>
> - Why PeerReview data is best combined with PULSE training (Responses to **W1-1**, item **2**)
>
> - Why PeerReview data has strengths over LM-judge (Responses to **W2-1**, item **1**)
>
> - Why Multi-agent setup is important (Responses to **W2-1**, item **2**)
>
> ---
>
> ***W4: I do think that this paper has potential to be impactful, if they can focus on how they designed the multi-agent system. First to see if just doing inference works on improve quality, then go to training .. it might be because of my preferred style of research is to understand 1 thing thoroughly compared to proposing many things together.***
>
> We certainly respect your style. Also, we hope you understand that the two contributions are essential to each other. We give a final summary of our contributions:
>
> - Our work is the first to propose a **multi-agentic paradigm of collaborating peer LLMs** to create preference data. We believe that our approach is a scalable way to address the recent issue of **data scarcity** in AI.
>
> - PeerReview data are constructed from **structured self-correction** guided by diversified critics.
>
> - PULSE training uses **robust loss combined with parameter scheduling** to mitigate potential noise or biases in the PeerReview data.
>
> - Their synergism is essential in outperforming existing baselines.
>
> We believe that two combined contributions are worth more than one to the community. We hope you could embrace our stance.
>
> ---
>
> ***Q1: Can the authors show the exact evaluation protocol for evaluation ? (Alpaca Eval decoding params, judge used)***
>
> Please refer to the response to **W3-2**.
>
> ---
>
> ***Q2: Can the authors explain the difference between their method (multi-agent peer review part) with using an LM-judge to evaluate quality?***
>
> Please refer to the responses to **W2-1**
>
> ---
>
> ***Q3: Use citet{} and citep{} correctly.***
>
> **Action:** Thank you for pointing this out. We reverted the citation style to the default ICLR style.

---

> > ### Comment · Reviewer_fsc5 · 2025-11-26
> > **Reviewer Response**
> >
> > First, I want to thank the authors for putting up such a comprehensive response. However, I still have a few concerns to address:
> >
> > > The PeerReview data achieves SoTA results for most of the benchmarks.
> >
> > From my understanding, SoTA (State of The Art) means that you have beat **all** past approaches. But this is clearly not true in that if you look at the official Alpaca Eval leaderboard (https://tatsu-lab.github.io/alpaca_eval/), there are many models that significantly outperform your trained model, and some of the models have the same size as your trained models. So I don't think you can claim SoTA. From my perspective, this work is more of a "proof-of-concept" work than an work that tries to be SoTA.
> >
> > However, a work does not need to be SoTA in order to be good and impactful, as long as you could demonstrate insights for the community. Therefore, my reviews does not take into account of whether you are SoTA. For example, I do like that you highlighted the need for **diversity** in feedback, the need for coping with the biases in LM reviewer responses (making them longer)
> >
> > > If the LLM over-fits lengthy responses in the early stages of training
> >
> > If the peer review process introduces spurious signals (e.g. length and other biases), I think it would be better to try to design a peer review system that has less of this bias (e.g. response filtering), instead of trying to mitigate it with auxiliary loss function. It is really hard to prove two things in a paper - first you would need to ablate the first part (peer review) with other synthetic data generation pipelines, and ablate the second part (PULSE) with different loss functions. I still insist that it is not absolutely necessary to put the two things together in one paper, and that keep the paper simple actually makes the paper stronger.
> >
> > > LM judge only ranks multiple candidate answers by relative quality.
> >
> > This is not correct I believe, as there are many work on LM-judges that aims to give feedback to a single answer (e.g. [1, 2]). I think asking an LM to give any sort of feedback on model generations constitutes as LM-judge.
> >
> > > it learns how to self-correct.
> >
> > My understanding of **self-correct** is when a model is able to refine responses **without** external feedback. However, since you are using an LM-reviewer, I don't think this constitutes as self-correct. Please consider using different language.
> >
> > ---
> > I decided to increase the overall rating (2 -> 4) and soundness score (3 -> 4), but I am unable to give a higher score due to the aforementioned reasons.
> >
> > ## References
> >
> > [1] Pairwise or Pointwise? Evaluating Feedback Protocols for Bias in LLM-Based Evaluation. Tripathi et al. COLM 2025
> >
> > [2] J1: Incentivizing Thinking in LLM-as-a-Judge via Reinforcement Learning. Whitehouse et al. 2025

---

> ### Author Response · Authors · 2025-11-28
> **Thank you for raising the score (2->4)**
>
> Thank you so much for raising the score and careful advice. Your detailed feedback has been an enormous help in improving our work. We will polish our work based on your advice. Below we will provide brief responses to your additional concerns; let us know if a more detailed rebuttal is needed.
>
>
> - **SoTA.** In your leaderboard, the highest-ranked model similar to our size (7B) is at \#18 **(29.7)**, and PULSE **(29.54)** is only 0.16\% behind that model. Considering heavy-tuning of leaderboard models, we believe PULSE performs quite well as a proof-of-concept. We did not use the term SoTA in the paper, but only in the rebuttal. We will change the term SoTA to *"outperforms existing baselines"*.
>
> - **Filtering**. In our work, we filtered the generated responses based on peer-review scores (discarded if scored $<3$) for quality control. We will additionally consider length-adjusted filtering as you suggested.
>
> - **LM-judge.** By "LM-judge" in the rebuttal, we meant *reward models*. As you mentioned, LM-judge means both providing scores or textual feedback; we meant the former in the context of our work.
>
> - **Self-correction.** We believe you mean internal self-correction in a stricter sense. There are works on self-correction with external signals, e.g., highly-cited ones in [1],[2]. We similarly consider such self-correction in a broader sense.
>
> **References**
>
> [1] Zelikman, Eric, et al. "STaR: Self-taught reasoner bootstrapping reasoning with reasoning." Proc. the 36th International Conference on Neural Information Processing Systems. Vol. 1126. 2024.
>
> [2] Shinn, Noah, et al. "Reflexion: Language agents with verbal reinforcement learning." Advances in Neural Information Processing Systems 36 (2023): 8634-8652.

---

### Official Review · Reviewer_Yi9x · 2025-11-01

**Soundness:** 3
**Presentation:** 2
**Contribution:** 3
**Rating:** 8
**Confidence:** 4

**Summary:**

This paper proposes a LLM post-training framework PULSE with two major contributions
* PULSE proposes a peer review and revision approach to synthesize preference data with better diversity.
First, an actor LLM generates initial responses, which are reviewed by multiple critic LLMs.
The actor LLM then revises the responses based on the feedback, and the revised responses are compared with the initial ones by the critic LLMs.
* PULSE proposes a loss function under the risk minimization framework to improve robustness.
The key theoretical tool is Equation 10, which shows that the loss function can be determined by the discriminate function, posterior distribution, and minimum conditional risk.
PULSE then follows the discriminate function and posterior distribution used in the Bradley-Terry model and derives the PULSE loss function by setting the minimum conditional risk as a quadratic function.

Experiments are conducted by training Mistral-7B, Llama-3-8B, Qwen2.5-7B, and Gemma-2-9B on UltraFeedback and evaluating on AlpacaEval, MT-Bench, and academic benchmarks, where PULSE demonstrates superior performance.
Ablation study illustrates the effects of designed components, *i.e.*, peer review and $\beta$-scheduling.

**Strengths:**

* It is common to synthesize data by revision in engineering.
The proposed method increases the training stability to noise in synthesized data, which is helpful to the community.

* It is interesting to derive the loss function under the risk minimization framework.
I carefully check the derivations of Equations 8 to 19 and found no issue.
Theoretical support enhances the persuasiveness of the proposed method and provides insights for future work.

* The experiment is comprehensive, which is conducted on four LLMs and yields consistent superior performance.
Ablation study is performed to illustrates the effects of designed components.

**Weaknesses:**

This is a generally high-quality paper and I did not find many flaws.
My main concern is that it seems there is no ablation on the PULSE loss.
Is it possible to achieve better performance by using Skywork RM to synthesizes preference data and PULSE loss to train the model?

**Questions:**

I see that under the risk minimization framework, PULSE and DPO losses share identical discriminant function and posterior distribution.
Why does using quadratic function as the minimal conditional risk lead to better stability?
Could the authors provide more theoretical analysis on this?
Additionally, can other preference losses be also explained under the risk minimization framework?
For example, are they equivalent to using different minimal conditional risks?
It would be beneficial to use a table for comparison.

---

> ### Author Response · Authors · 2025-11-26
> **Response to Reviewer Yi9x (1/2)**
>
> We sincerely appreciate the reviewer for valuable comments which are detailed, constructive, and to the point. We have addressed all of the concerns raised by the reviewer.
>
> ---
>
> ***W1: This is a generally high-quality paper and I did not find many flaws. My main concern is that it seems there is no ablation on the PULSE loss. Is it possible to achieve better performance by using Skywork RM to synthesize preference data and PULSE loss to train the model?***
>
> Thank you very much for your positive evaluation.
>
> **1. Applying PULSE loss to existing preference datasets.**
>
> We provided an ablation on the PULSE loss in the original paper: in Table 7 of Appendix A.7, we provided small experiments on standard preference dataset (UltraFeedback) combined with PULSE loss. In the revised paper, we will provide **expanded ablations** on existing preference datasets.
>
> Table 1 shows the expanded results of applying PULSE training to existing preference datasets: UltraFeedback [1] and HH-RLHF [2]. Notably, using UltraFeedback with PULSE training yields a modest $+4.38$ improvement (17.11 $\rightarrow$ 21.49) from the baseline. In contrast,  combining PULSE training with PeerReview data achieves **29.54**, an additional $+8.05$ gain. The PeerReview data achieves SoTA results for most of the benchmarks.
>
> | Method                  | Alpaca LC. | Alpaca 2.0 | GSM8K | MATH  | MBPP  | MMLU  |
> |------------------------|------------|------------|-------|-------|-------|-------|
> | Mistral-7B             | 17.11      | 14.72      | 44.50 | 10.02 | 14.20 | 55.96 |
> | PULSE w/ Ultrafeedback | 21.49      | 19.67      | 44.05 | 10.56 | 34.03 | 54.42 |
> | PULSE w/ HH-RLHF       | 13.61      | 16.59      | 46.07 | 12.84 | 32.00 | 53.61 |
> | PULSE w/ PeerReview    | **29.54**  | **30.24**  | **47.61** | **12.70** | **37.80** | **57.53** |
>
> Table 1.
>
> The high performance of the PeerReview data can be attributed to **self-correction**. The actor LLM generates and refines its responses based on the feedback from diverse peer critics. This improves the reasoning abilities of the actor. Moreover, since the data are self-generated, they **better match the output distribution** of the actor LLM, which can facilitate preference training.
>
> **2. Applying PULSE loss to Skywork RM.** Per your suggestion, we experimented with applying PULSE training directly to the dataset based on  Skywork RM. The results are shown in Table 2.
>
> | Method              | Alpaca LC. | GSM8K | MATH  | HumanEval | MBPP   | MMLU  |
> |---------------------|------------|-------|-------|-----------|--------|-------|
> | DPO w/ Skywork RM   | 27.45      | 45.13 | 10.25 | 8.72      | 19.24  | 55.62 |
> | PULSE w/ Skywork RM | **28.59**  | **46.15** | **10.68** | **10.56** | **21.78** | **56.56** |
>
> Table 2.
>
> **Conclusion.** On both the UltraFeedback dataset and the data synthesized by Skywork RM, PULSE training consistently outperforms DPO. The results confirm that PULSE training is a more robust and generalizable learning algorithm.
>
> **Action:** We added the comparisons with existing preference datasets to Experiments section (Table 6) and results of Skywork Appendix A.7 in the revised paper.
>
> **References**
>
> [1] Cui, Ganqu, et al. "Ultrafeedback: Boosting language models with high-quality feedback." (2023).
>
> [2] Bai, Yuntao, et al. "Training a helpful and harmless assistant with reinforcement learning from human feedback." arXiv preprint arXiv:2204.05862 (2022).

---

> ### Author Response · Authors · 2025-11-26
> **Response to Reviewer Yi9x (2/2)**
>
> ***Q1: I see that under the risk minimization framework, PULSE and DPO losses share identical discriminant function and posterior distribution. Why does using quadratic function as the minimal conditional risk lead to better stability? Could the authors provide more theoretical analysis on this? Additionally, can other preference losses be also explained under the risk minimization framework? For example, are they equivalent to using different minimal conditional risks? It would be beneficial to use a table for comparison.***
>
> **1. Robustness of PULSE.**
>
> Indeed, DPO and PULSE share the same discriminant function $f(x)$ and posterior model $p(f)$; they differ only in the choice of minimum conditional risk $C(p)$ and the resulting loss function $\phi$. The following is a comparison of the losses:
>
> - **DPO (Binary Entropy):** $\phi_{\text{DPO}}(f) = -\log \sigma(f)$.
> - **IPO (Quadratic Risk):**, $\phi_{\text{PULSE}}(f) = (1-\sigma(f))^2$ (Brier score / MSE).
>
> We compare the behavior of two losses:
>
> - **DPO loss is unbounded, PULSE loss is bounded.** Consider a severely misclassified sample. This is the regime where $f \to -\infty$. In this case
> $$
> \phi_{\text{DPO}}(f) = -\log\left(\frac{1}{1+e^{-f}}\right) \approx -\log\left(\frac{1}{e^{-f}}\right) = -f
> \to \infty \quad \text{(unbounded)}
> $$
> and
> $$\phi_{\text{PULSE}}(f) = \left(\frac{1}{1+e^f}\right)^2 \to 1 \quad \textrm{(bounded)}$$
> The boundedness of PULSE loss helps the stability of training.
>
> - **PULSE loss has decreasing gradient.**
> We have that
> $$\phi_\text{PULSE}'(f) = \frac{-2e^f}{(1+e^f)^3}$$
> and
> $$\phi_\text{PULSE}''(f) = \frac{6e^f}{(1+e^f)^4} - \frac{2e^f}{(1+e^f)^3}$$
>
> By setting $\phi_\text{PULSE}''(f) = 0$, we get $f= -\log 2$. This means that $\phi_\text{PULSE}(f)$ is a concave function of $f$ for $f<-\log 2$. This implies that the gradient $\phi_\text{PULSE}'(f)$ is **{decreasing** in $f$ for $f<-\log 2$.
>
> The decreasing gradient make the training robust to noise. Suppose there is "spike" in $f$ due to noise. In other words, suppose there is negative noise $-\epsilon$ and $f$ is corrupted to $f-\epsilon$. The gradient of loss, which is the update of the model, actually **decreases** from $\phi_\text{PULSE}'(f)$ to $\phi_\text{PULSE}'(f-\epsilon)$. This means that the update of the model due to noise actually decreases; this enables a noise-robust training under PULSE. In contrast, one can show that  $\phi_\textrm{DPO}(f)$ is convex in $f$; meaning that $f$ has nondecreasing gradient.
>
> **2. Other preference losses and risk minimization.** We searched for other preference losses to which we can apply risk minimization.
>
> From recent works, we found the losses of IPO [1]
>
> $\mathcal{L}_\textrm{IPO}=(\delta-(r_w-r_l))^2$
>
> and SLiC [2]
>
> $\mathcal{L}_\textrm{SLiC}=\max(0, \delta-(r_w-r_l))$
>
> These losses have the following in common: $r_w$ denotes the reward for the winning response and $r_l$ denotes the reward for the losing response. Now we can apply the risk minimization framework from [3] to find the corresponding losses and minimum conditional risks. We provide a summary of the comparison of these losses in Table 3.
>
> | Loss | $C(p)$ | $\phi(f)$ | Key Property $(f \to -\infty)$ |
> |------|----------|-------------|-----------------------------------|
> | DPO  | $-p \log p - (1-p)\log(1-p)$ | $-\log \sigma(f)$ | Unbounded, nondecreasing gradient, sensitive to noise |
> | IPO [1] | $p(1-p)$ | $(1-f)^2$ | Unbounded, nondecreasing gradient, sensitive to noise |
> | SLiC [2] | $1 - \lvert 2p - 1\rvert$               | $\max(1 - f, 0)$             | Unbounded, nondecreasing gradient, sensitive to noise                      |
> | PULSE | $p(1-p)$ | $(1 - \sigma(f))^2$ | Bounded, decreasing gradient, robust to noise |
>
> Table 3.
>
> **Action:** We add discussion and Table 3 in the Appendix A.9 of the revision.

---

> ### Comment · Reviewer_Yi9x · 2025-11-26
>
> I thank the authors for the response.
> I think this paper is interesting and contributing.
> I see all other reviews are negative.
> If this paper is rejected, I encourage the authors to refine the presentation and submit to the next venue.

---

> ### Author Response · Authors · 2025-11-27
> **Thank you for the positive feedback**
>
> Thank you so much for positive and encouraging comments. We will also try our best with our rebuttal to address other concerns.

---

### Author Response · Authors · 2025-12-02
**General Response and Revision Summary**

Dear AC and Reviewers,

Thank you for your service during this difficult time. Below we provide a summary of score updates and revisions.

**1.Score Updates**

**- Summary**

- **Initial Scores:** 8, 4, 4, 2  (avg. 4.5)

- **Post-discussion Scores:** **8, 6, 6, 4  (avg. 6.0)**

**- Score Increases**

**Reviewer Yi9x (8 — Champion):** Maintained score of 8 and stated: *"I think this paper is interesting and contributing…"*

**Reviewer YF36 (4 $\rightarrow$ 6):**
On Nov 27, the reviewer stated: *"I am willing to raise my score from 4 to 6 :)"* This was motivated by our experiments separating the effects of PeerReview data from the PULSE loss.

**Reviewer fsc5 (2 $\rightarrow$ 4):**
On Nov 27, the reviewer stated: *"I decided to increase overall rating (2 → 4)…"* The reviewer acknowledged our comprehensive response and the importance of our insights on multi-agent feedback diversity.

**Reviewer 9BAx (4 $\rightarrow$ 6):**
On Nov 28, the reviewer stated: *"I would be inclined to raise my score to 6, but it seems that I am unable to do so due to an accident. I leave my tendency here as a reference for AC."* ”The score increase followed our clarification of necessity of PeerReview data and demonstrated performance gains (Table 2 in the rebuttal).

**2. Summary of Strengths**

We thank reviewers for encouraging comments including:

- **Novelty and Methodological Impact:**

"The main idea of this paper: multi-agent collaborating to give and refine a response, is interesting and novel."

"The iterative review–revise–learn structure is an elegant generalization of preference alignment: it moves beyond one-shot comparisons toward process-level refinement."

"The proposed method increases the training stability to noise in synthesized data, which is helpful to the community."

- **Theoretical Soundness:**

"The methodology is technically sound and well-motivated. ... and the connection to policy improvement and reward regularization is mathematically justified."

"It is interesting to derive the loss function under the risk minimization framework. I carefully check derivations ... and found no issue."

"Theoretical support enhances the persuasiveness of the proposed method and provides insights for future work."

- **Comprehensive Experiments and Evaluation:**

"The experiment is comprehensive, which is conducted on four LLMs and yields consistent superior performance."

"There are multiple benchmarks selected from difference domains to evaluate the model."

"Ablation study is performed to illustrate effects of designed components."

- **Clarity and Organization:**

"The paper is written clearly to understand."

"The paper is well organized."

**3. Summary of Key Improvements and Revisions**

The following are major updates made in the paper to thoroughly address the reviewers’ concerns.

- **Generalizability (9BAx, YF36, fsc5):**

Applied PULSE training to standard preference datasets (UltraFeedback, HH-RLHF) and Skywork-RM.

**Result:** PULSE consistently outperformed DPO, and the combination **PULSE + PeerReview Data** achieved the strongest results.

- **Ablation Studies (YF36, fsc5):**

Added ablation studies on Critic Diversity (Self-critic vs. Single-critic vs. Multi-critic) and Textual Feedback.

**Result:** Removing critic diversity or textual feedback caused large performance drops.

- **Clarified Contributions:**

Refined the explanation of how PeerReview Data (self-correction) and PULSE loss (robustness) bring synergistic effects to achieve SOTA results.

- **Generalizability \& Data Efficiency (9BAx, YF36, fsc5)**

**Action**: Added experiments on UltraFeedback, HH-RLHF, and Skywork-RM.

**Result**: PULSE consistently surpassed DPO across all datasets.
Moreover, **PULSE + PeerReview Data** delivered the highest performance, validating our multi-agent data design.
(Added to Table 6 and Section 313)

- **Necessity of Multi-Agent Framework (YF36, fsc5)**

**Action**: Added ablations for critic diversity and textual feedback.

**Result**: Showed large performance drops without multi-agent critics or structured feedback, showing that simple LM-judges cannot replicate our collaborative pipeline.
(Added to Table 5 & Appendix A.6)

- **Theoretical Grounding of Robustness (Yi9x)**

**Action**: Added a risk-minimization analysis comparing PULSE gradients with various preference optimization methods, e.g., DPO, IPO, and SLiC.

**Result**: Derived that PULSE has a *decreasing gradient property*, formally explaining its robustness to noise.
(Added to Appendix A.9)

- **Reproducibility Improvements (fsc5)**

**Action**: Fully documented evaluation protocols, including judge model (*GPT-4-1106-preview*) and decoding settings for AlpacaEval.
(Added to Appendix A.12)

**4. Conclusion**

We thank the reviewers for their acknowledgment and positive feedback. We also thank the new AC for the extra service. We hope you could consider the score increase resulted from active participation of reviewers.

Best regards,

Authors

---

### Note · Authors · 2026-01-26

I have read and agree with the venue's withdrawal policy on behalf of myself and my co-authors.

---

### Meta-Review · Area_Chair_oXHX · 2025-12-29

**Summary:**

The paper introduces PULSE, a framework designed to synthesize preference data through a multi-agent "peer-review" process where LLMs critique and refine each other’s outputs. Additionally, the authors propose a noise-robust quadratic loss function intended to improve preference learning from potentially imperfect synthetic data. Despite a comprehensive rebuttal and some improvements to the manuscript, the submission still lacks a focused contribution that it integrates several part all together without identifying key aspects to address for self-correction.

**Reviewer Concerns:**

SOTA: Reviewer fsc5 noted that on the Alpaca Eval leaderboard, several models of comparable size (e.g., 7B parameter models) consistently outperform PULSE.

Incremental contributions: Reviewers YF36 & fsc5 commented that the data synthesis and the robust loss were two separate topics that should have been studied more independently. While the authors provided individual ablations, the reviewer's concern on specific in-depth study of each topic remain unaddressed.

**Reviewer Scores:**

Reviewer YF36  from 4 to 6
Reviewer fsc5 from 2 to 4
Reviewer 9BAx from 4 to 6

---

### Decision · Program_Chairs · 2026-01-26

Reject